# Are lipids always depleted? Comparison of hydrogen, carbon, and nitrogen isotopic values in the muscle and lipid of larval lampreys

Thomas M. Evans ⬡¤*, Shale Beharie

Biology Department, St. Mary's College of Maryland, St. Mary's City, Maryland, United States of America

¤ Current address: Department of Natural Resources and the Environment, Cornell University, Ithaca, New York, United States of America
* thomas.mark.evans@gmail.com

**Data Availability Statement:** All relevant data are within the manuscript and its Supporting Information files.

## Abstract

Stable isotope ratios in organisms can be used to estimate dietary source contributions, but lipids must first be accounted for to interpret values meaningfully. Lipids are depleted in heavy isotopes because during lipid synthesis light isotopes of carbon ($^{12}C$) and hydrogen ($^{1}H$) are preferentially incorporated. Prior work in larval lampreys has noted unusual lipid effects, which suggest lipids are enriched in the heavy isotope of carbon ($^{13}C$), but still depleted in the heavy isotope of hydrogen (deuterium; $^{2}H$); nitrogen, a relatively rare element in lipids, has not been identified as being as sensitive to lipid content. Our objective was to determine if stable isotope ratios of hydrogen, carbon, and nitrogen behaved as expected in larval lampreys, or if their lipids presented different isotopic behavior. The $\delta^{2}H$, $\delta^{13}C$, and $\delta^{15}N$ were measured from the muscle of four lamprey species before and after lipid extraction. In addition, muscle of least brook lamprey (*Lampetra aepyptera*) was collected every three months for a year from two streams in Maryland. Isotopic ratios were measured in bulk and lipid-extracted muscles, as well as in extracted lipids. The difference between muscle samples before and after lipid extraction ($\Delta\delta^{2}H$, $\Delta\delta^{13}C$, $\Delta\delta^{15}N$) was positively related to lipid proxy (%H or C:N ratio) and were fit best by linear models for $\Delta\delta^{2}H$ and $\Delta\delta^{15}N$, and by a non-linear model for $\Delta\delta^{13}C$. The difference between lipid-extracted muscle and lipid $\delta^{13}C$ ($\Delta_{ML}\delta^{13}C$) was negative and varied between months (ANOVA, $F_{3,53} = 5.05$, p < 0.005). Our work suggests that while lipids are often depleted in $^{13}C$, this is not a universal rule; however, the depletion of $^{2}H$ in lipid synthesis appears broadly true.

## Introduction

Stable isotope ratios are a powerful tool which have been used to deepen our understanding of numerous biological processes including physiology, nutrient cycling, food web structure, diet, and migration [1]. In natural systems, elements generally have one to six stable isotopes,

**Funding:** The author(s) received no specific funding for this work.

**Competing interests:** The authors have declared that no competing interests exist.

but usually two predominate, a light isotope that is common (>95% of all atoms) and a rare heavier isotope [2]. The ratio of the heavy to the light isotopes varies between organisms and their food sources because lighter isotopes are favored during enzymatic reactions [2, 3]. Lighter isotopes have a lower activation energy, and therefore proceed more rapidly when reacting than heavy isotopes [2, 4]. The resulting difference between an organism and its dietary sources (i.e., discrimination, although sometimes called fractionation [3]) has been the subject of much study [5–7].

An organism's stable isotope signature not only reflects materials incorporated from its diet, but also isotope discrimination during the generation of new biomolecules [8]. For instance, while food sources can be used to build cellular components (e.g., proteins), they can also provide resources to synthesize lipids. Enzymatic synthesis of lipids favors lighter isotopes at a high rate; for carbon this occurs when pyruvate is converted to acetyl-CoA [9] and for hydrogen it occurs at various steps [10]. As a result, lipids are depleted (i.e., they contain lower amounts than the starting materials) in the heavy isotopes of carbon ($^{13}$C) and hydrogen ($^{2}$H), and higher amounts of the light isotopes of carbon ($^{12}$C) [6, 9, 11] and hydrogen ($^{1}$H) [10, 12]. In contrast, nitrogen isotopes, which are relatively rare in lipids, are incorporated at approximately the same rates during lipid synthesis [11].

Lipids are ubiquitous in living organisms, and researchers using carbon and hydrogen stable isotope ratios need to account for lipids to make meaningful interpretations [6, 13] either by chemically extracting them from a sample before stable isotope analysis or mathematically correcting for them *a posteriori* using a validated normalization model [6, 13–15]. Extraction of lipids from samples increases the cost of the study and the number of samples for analysis [14, 16], as lipid extraction can remove lipophilic proteins which could alter nitrogen stable isotope ratios [14]. If lipid extracting samples, the best practice is to run the lipid-extracted samples for carbon and hydrogen analysis and the bulk samples for nitrogen analysis [6]. Therefore, workers often prefer to mathematically correct samples with validated models instead, using a measure (direct or proxy) of the lipid content in bulk samples and estimating a lipid-free value.

Stable isotopes of carbon ($\delta^{13}$C) are often measured simultaneously to stable isotopes of nitrogen ($\delta^{15}$N) and the ratio of C:N routinely reported by stable isotope laboratories. The C:N ratio of animal samples can be used as a proxy for lipid content because the C:N ratio increases as carbon is sequestered into lipids [6, 11, 17]. Therefore, as C:N values increase the $\delta^{13}$C values in tissues decreases (because of the aforementioned reductions in lipid $^{13}$C); this phenomenon has been demonstrated in a wide-range of animals [6, 11]. However, in numerous studies of larval lampreys, authors have reported that as C:N ratios increased, $\delta^{13}$C also increased [18–20]. Limited work has been done with larval lamprey $\delta^{2}$H [21–23], but when bulk samples are measured, $\delta^{2}$H declined as C:N ratios increased [23]. These relationships (correlation between $\delta^{13}$C or $\delta^{2}$H and C:N ratios) suggest that as lipids rise in larval lampreys, bulk isotopic values are increasingly influenced by lipids.

Lampreys are jawless fishes, which belong to the most basal extant vertebrate group [24], with a complex life cycle including a larval period and a true metamorphosis [25]. Although there are ~40 species of lampreys, all species have a similar larval period. The larval period usually lasts from 3–7 years (although it can be shorter or longer) and is used to collect sufficient lipid reserves to fuel metamorphosis [26]. Larval lampreys are primary consumers which feed by collecting detritus (e.g., decaying organic matter and bacteria) [27, 28], often predominantly allochthonous in origin (i.e., terrestrial) [21, 29], but to varying extents incorporate autochthonous algae from stream sediments [26]. Larvae grow slowly, often reaching only a few grams in weight over the entire larval period [30], and accumulate large quantities of lipids in their bodies (~15% of their wet weight) [31, 32]. Larval growth is not constant; larvae grow most

during the spring and to a lesser extent the fall [33], often declining in condition throughout the summer [31, 34]. After the multi-year larval period, lampreys undergo a metamorphosis at the end of the summer during which they do not feed, instead they rely on their accumulated lipid stores [26, 32].

Isotope ratios in larval lamprey muscle tissue are often highly variable [18–20, 22, 35], although many studies have not accounted for lipids [18–20, 35]. However, if larval lamprey muscle $\delta^{13}C$ is mathematically corrected for lipids following published equations [6, 11], larval lamprey $\delta^{13}C$ rises and becomes unexplainably enriched [18]. Interestingly, although more limited work on larval lampreys using $\delta^2H$ has been done [21, 22], when lipids were extracted $\delta^2H$ values have risen [21], as would be expected based on theoretical expectations. Prior work has variably interpreted stable isotopes in larvae, either as larvae consuming a range of food sources but often incorporating large quantities of algae [18, 19] or, if lipids were removed, as larvae being almost entirely dependent on allochthonous detritus [21, 22]. Larval lamprey lipids may be unusual and contributing to these remarkable signatures, but this needs to be formally established in the literature body through measurements of stable isotope ratios in both the muscle and the lipid.

To ensure larval lamprey $\delta^2H$ and $\delta^{13}C$ are useful in predicting source dependence it is necessary to identify if their lipid stable isotopes can deviate from expectations, but more broadly because lipids are a common concern in stable isotope analysis and lampreys may suggest alternative isotope portioning during lipid synthesis. Additionally, the use of $\delta^2H$ in stable isotope ecology is growing, and recommendations for interpreting their values are increasingly important to allow broader use [36]. Therefore, the objectives of this study were: 1) to establish if larval lamprey muscle tissue conformed to expectations–we hypothesized $\delta^2H$ would be enriched in muscle after lipid extraction following literature precedent [10], muscle would be depleted in $\delta^{13}C$ after lipid extraction based on prior observations that bulk muscle $\delta^{13}C$ increased with C:N ratio [18, 19], and $\delta^{15}N$ would remain unchanged; 2) to evaluate the type of relationship between the isotopic ratio and lipid proxy; 3) determine if the difference in stable isotope ratios between lipid-extracted muscle and lipids was consistent throughout the year.

## Materials and methods

### Larval lamprey samples

Larval least brook lamprey (*Lampetra aepyptera*) were sampled from two rivers (Henderson Run and Furnace Brook) near St. Mary's City, Maryland in September and December of 2020, and then in March and May of 2021 (Fig 1). Larvae generally live in soft sediments and depositional areas [26]. Larvae were collected alive from these habitats along a stream reach of 100 m by backpack electrofisher (LR-24, Smith Root) using two power settings. Larvae were encouraged to leave the sediment with a low-power setting (3 hz, 100 V, 2:2 pulse pattern), and immobilized for collection with a high-power setting (30 Hz, 120 V, continuous output) when they were observed. Sea lamprey larvae were collected once from the Little Patuxent River (Maryland, USA) in May 2021, and least brook lamprey from Greenhill Run in September 2020 (Maryland, USA). Within two hours of collection, larvae were transported in stream water to the laboratory. Larvae were immediately euthanized by overdose in MS-222 and then dissected, or if time was insufficient to process all euthanized larvae rapidly, they were frozen at -20°C until they could be processed later. To dissect the larvae >50 mm, euthanized larvae were filleted by cutting behind the gills and removing a fillet along the left side of the body. The skin was then peeled from the body and the remaining material was considered muscle. If the larvae were <50 mm, the head and the visceral contents were removed before the

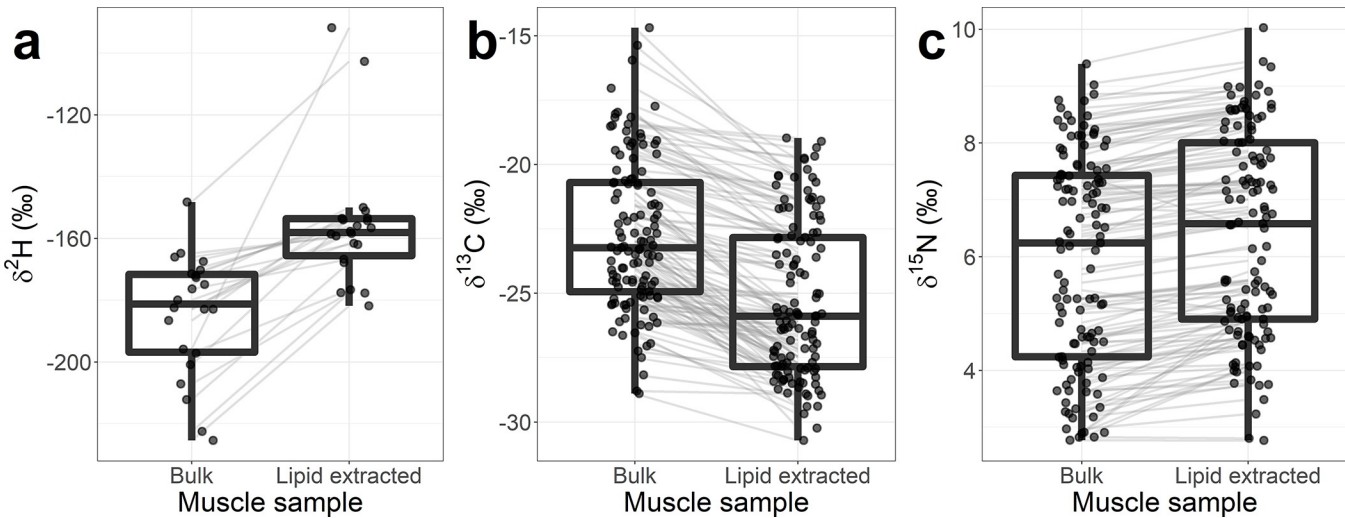

**Fig 1. Muscle sample a) δ²H, b) δ¹³C, and c) δ¹⁵N before and after lipid extraction.** Boxplot midlines are the median, the boxes enclose the 1ˢᵗ-3ʳᵈ quartiles (i.e., 50% of the data points), and the whiskers are 1.5 times this interquartile range. Measured values that boxplots are drawn from are represented as points and are jittered to make them more easily distinguishable. The difference before and after extraction for each paired value is shown as a light grey line between groups.

notochord was pulled from the body and the skin peeled off; the remaining material was considered a muscle sample. The collection and handling of these lampreys was approved by St. Mary's College of Maryland institutional animal care and use committee (IACUC, Permit number: SCP202090).

Muscle samples were placed into a microcentrifuge tube and then dried at 60°C for 24 hours in a drying oven. After drying, muscle samples were homogenized in the microcentrifuge with a metal rod, which was cleaned between each sample. A subsample was lipid extracted following a modified Folch method [37]. Briefly, subsamples were washed with ~1 mL of 2:1, by volume, chloroform to methanol three times for ~1 minute each time at room temperature. Samples were spun in a centrifuge at 5000 rpm for five minutes between each wash before the supernatant was decanted and more chloroform methanol was added. All supernatants were collected and dried for 24 hours at 60°C in a drying oven. After three washes the muscle was considered lipid free, as the supernatant was clear, and the muscle had become brittle and lighter in color. Lipid-free muscle was then ground with a metal rod in the same centrifuge tube. The container with evaporated supernatant had a viscous yellow to orange liquid, assumed to be lipid.

To increase the number of muscle samples, data from larvae collected from other locations were also used in this study. Least brook lamprey from Clearfork River (Ohio, USA) and American brook lamprey (*Lethenteron appendix*) from the Mad River (Ohio, USA) were collected in 2010. The collection and handling of these lampreys was approved by The Ohio State University IACUC. The δ¹³C and δ¹⁵N from these lampreys have already been published, but δ²H was also measured in these samples and was only included in a thesis [23]. Additionally, American brook lamprey collected in the Genesee River in August 2017 were also used for analysis in this paper. Again, muscle δ²H, δ¹³C, δ¹⁵N were previously published [21], but the study only reported lipid-extracted muscle or muscle with relatively low C:N values (C:N < 4). Lastly, Pacific lamprey (*Entosphenus tridentatus*) larvae from [38], which were tank reared their entire lives in Washington, USA, were also included in the analyses.

The way in which those larvae were collected is reported in full in each study, but a short summary is provided here. Wild caught larvae were collected using a backpack electrofisher

with two shock settings. All larvae were euthanized by overdose in MS-222 or clove oil and then dissected by filleting and skinning lateral muscle, being careful to avoid any organs or the notochord. Muscle samples were dried and stored as for muscle samples collected in Maryland. If lipids were extracted, they were removed following the modified Folch method described above but these studies did not retain the lipids. Lipids are rarely measured in stable isotope studies examining food webs, because they are mostly a nuisance factor which biases interpretation of trophic linkages [6]; only stable isotope ratios from Maryland larval lipid samples were measured.

Stable isotope ratios are reported in $\delta$ (delta) notation using the equation: $\delta^x Y = [(R_{sample}/R_{standard}) - 1] * 1000$, where Y is the element, x is the heavy isotope measured, and R is the ratio of heavy to light isotope in the sample and universal standards. As a result, isotope ratios are reported in per mil (‰) and can be positive or negative. Positive values mean the sample is enriched in the heavy isotope, while negative values mean the sample is depleted in the heavy isotope relative to the standard.

Homogenized samples for $\delta^{13}C$ and $\delta^{15}N$ were weighed (0.2–1.2 mg) and packed into tin capsules, while those for $\delta^2H$ (sometimes denoted as $\delta D$; 0.19–0.38 mg) were packed in silver capsules. Lamprey samples collected in Maryland were sent to the Cornell University Stable Isotope Laboratory (COIL). At COIL, samples were analyzed for $\delta^{13}C$ and $\delta^{15}N$ on a Thermo Finnigan Delta Plus and $\delta^2H$ samples were analyzed on a Delta V IRMS through a ConFlo III. Standard deviations for 3–4 internal standards (5–8 standard samples for each type per run) were 0.2‰ for $\delta^{13}C$, 0.3‰ for $\delta^{15}N$, 2‰ for $\delta^2H$. Standards at COIL for $\delta^2H$ were: 1) benzoic acid, 2) caribou hoof standard, 3) kudu horn standard, and 4) keratin. Samples from prior studies were either analyzed by COIL on the same instruments or at the University of California (UC) Davis Stable Isotope Facility (SIF) using a PDZ Europa ANCA-GSL (EA) attached to a PDZ Europa 20-20 isotope ratio mass spectrometer (IRMS). We did not send replicate samples between laboratories to test the reliability of measurements on different instruments, but internal standards had similar deviations, and we assumed that differences would be small and contribute little, if at all, to interpretive differences.

## Statistical analysis

To test if lipid-extraction changed the stable isotope ratio of sample, a separate paired t-test for each isotope ratio was used to compare bulk muscle samples ($\delta^x Y_{bulk}$) and lipid-extracted muscle ($\delta^x Y_{extracted}$). Following this, we anticipated that magnitude of change after extraction would increase with the lipid content of a sample. We used C:N ratios as a lipid proxy for $\delta^{13}C/\delta^{15}N$ samples based on literature precedent [6, 17]. However, C:N ratios are not directly measured on $\delta^2H$ subsamples but are instead measured on a paired $\delta^{13}C/\delta^{15}N$ subsample, offering an opportunity for increased variability. As a result, we explored whether %H, which is measured on $\delta^2H$ subsamples, might be utilized instead. We regressed $\delta^2H_{bulk}$ against both C:N and %H with simple linear models. We also regressed %H against C:N with a linear model. We compared significant terms and $R^2$ values to determine how well these models fit.

We regressed lipid proxies (%H and C:N ratios) against differences stable isotope ratios in bulk muscle and lipid extracted muscle. The difference between bulk muscle and lipid-extracted muscle was calculated as follows: $\Delta\delta^2H = \delta^2H_{extracted} - \delta^2H_{bulk}$, $\Delta\delta^{13}C = \delta^{13}C_{extracted} - \delta^{13}C_{bulk}$, and $\Delta\delta^{15}N = \delta^{15}N_{extracted} - \delta^{15}N_{bulk}$. We fit the $\Delta\delta^{13}C$ and $\Delta\delta^{15}N$ against the C:N ratio (a common proxy for lipid content [6, 11]) with four models, all of which have been used to fit similar data in prior studies [6, 11]. Our simplest model was a linear model (fit by the lm function in R):

$$\Delta\delta^x Y = a \cdot C:N + b \qquad (1)$$

where b is the y-intercept and therefore -b/a is the estimate of C:N in a lipid-free sample (x-intercept). We also ln-transformed the C:N data [11] (fit by the nls function in R):

$$\Delta\delta^x Y = a \cdot \ln(C:N) + b \qquad (2)$$

and as a result, $e^{(-a/b)}$ would be the model estimate of C:N in a lipid-free sample.

We also fit the data with a non-linear model [11] (fit by the nls function in R):

$$\Delta\delta^x Y = \frac{(a \cdot C:N + b)}{(C:N + c)} \qquad (3)$$

where a is an asymptote (i.e., an estimate of the difference between lipid and muscle), -b/a is the estimate of C:N in a lipid-free sample (x-intercept). Lastly, we also fit a simplified version of Eq 3 (fit by the nls function in R):

$$\Delta\delta^x Y = \frac{(a \cdot C:N + b)}{C:N} \qquad (4)$$

where a remains the asymptote and -b/a is the estimate of C:N in a lipid-free sample. We examined model residuals to test for homoscedasticity and compared models using Akaike Information Criterion (AIC) values using the AIC function in R v4.2.1. Mean square error (MSE), mean absolute error (MAE) were also calculated to understand model fit to the data. If models had similar support based on AIC scores (AIC scores < 4 units different), MSE, and MAE, the simpler model was selected as the top model. Samples from all sites in Maryland and the Genesee River (New York) [21] contributed to this analysis.

For $\Delta\delta^2 H$ we used the Eqs 1, 2 and 4, but we used the percentage of hydrogen (%H) in the untreated sample instead of C:N ratio. We chose to use percent hydrogen because this was measured on each $\delta^2 H$ sample (C:N was measured on a separate subsample submitted for $\delta^{13}C/\delta^{15}N$). The percentage hydrogen in the body should increase with lipids [39] since lipids include numerous hydrogens, the other cellular components (e.g., proteins and carbohydrates) were not expected to increase. Model selection followed the same pattern as described for $\delta^{13}C/\delta^{15}N$. Samples from Maryland (Henderson Creek and Johns Creek), the Genesee River (New York) [21], and tank reared larvae [38] contributed to this analysis.

We did not include species, location, or date of collection in $\Delta\delta^x Y$ models. Larval lampreys all have similar life history and are not expected to synthesize lipids using different biochemical pathways, regardless of species or where they are collected. We anticipated enzymatic discrimination during lipid synthesis would be similar year-round because there is little published stable isotope data for larval lamprey outside of the summer growing season.

To test model efficiency, we used three approaches: 1) the number of samples within a machine uncertainty (MU) window of the model fit, 2) the ability to estimate %H or C:N in a lipid free sample, and 3) the estimated maximum difference between bulk muscle ($\delta^x Y_{bulk}$) and lipid stable isotope ratios ($\delta^x Y_{lipid}$). We calculated the proportion of values within twice the standard deviation of the isotope ratio standards (4‰ for $\delta^2 H$, 0.4‰ for $\delta^{13}C$, 0.6‰ for $\delta^{15}N$) using the following: $| (\Delta\delta^x Y_{predicted} - \Delta\delta^x Y_{observed}) | > MU$. For the second approach, we assumed that after lipid extraction muscles were lipid-free, so we used a t-test to compare the top $\Delta^x Y$ model estimate of the percentage hydrogen in a lipid-free sample or C:N in lipid-free sample with those measured in lipid-extracted muscle. For the third, we calculated the difference between muscle and lipid samples ($\Delta_{ML}\delta^2 H = \delta^2 H_{extracted} - \delta^2 H_{lipid}$ and $\Delta_{ML}\delta^{13}C = \delta^{13}C_{extracted} - \delta^{13}C_{lipid}$) and comparing it to the asymptote (if present) in the top model.

There was evidence that $\Delta_{ML}\delta^x Y$ varied between collection months. Therefore, to test if variation in $\Delta_{ML}\delta^2 H$ or $\Delta_{ML}\delta^{13}C$ was larger in some months than others, two analyses of variance

(ANOVA) with $\Delta_{ML}\delta^2H$ or $\Delta_{ML}\delta^{13}C$ as the response and month as a predictor were used; a Tukey's honest significant difference test was used to compare contrasts. All analyses were performed in R v4.2.1 [40].

## Results

### Lipid extraction effect on isotope ratios

All three paired t-tests showed that muscle isotope ratios before and after lipid extraction were significantly different (Fig 1). After extraction, the $\delta^2H$ of muscle rose 29‰ ($T_{21}$ = -6.25, p-value < 0.0001), $\delta^{13}C$ declined 2.5‰ ($T_{120}$ = -16.5, p-value < 0.0001), and $\delta^{15}N$ increased 0.45‰ ($T_{120}$ = 12.4, p-value < 0.0001). The $\delta^2H$ and $\delta^{13}C$ changed in the directions supportive of our initial hypotheses. We did not hypothesize $\delta^{15}N$ would change, but it increased after lipid extraction.

Since lipid extraction altered all stable isotope ratios (Fig 1), we regressed stable isotope ratios of bulk muscle against the C:N ratio in a sample. For $\delta^2H$, the range of C:N ratios in bulk muscle was large (range: 3.5–10.0; Fig 2A). The $\delta^2H_{bulk}$ was negatively correlated with C:N ratio in a sample ($F_{1,75}$ = 46.45, p-value < 0.001, ρ = -0.62; Fig 2A). The percent hydrogen was positively correlated with the C:N ratio ($F_{1,75}$ = 98.65, p-value < 0.001, ρ = 0.75), although variation increased as C:N ratios increased (Fig 2B). Muscle $\delta^2H_{bulk}$ was negatively correlated with the percent hydrogen of a sample ($F_{1,75}$ = 219.1, p-value < 0.001, ρ = -0.86; Fig 2C). This linear model ($\delta^2H$ vs. %H) had a higher $R^2$ (0.75) and a lower residual standard error (RSE = 11.8) than the $\delta^2H$ vs C:N ratio linear model ($R^2$ = 0.38, RSE = 18.4). Although the slope term for the percent hydrogen vs. C:N ratio linear model was significant ($T_{1,75}$ = -14.8, p-value < 0.001) the intercept term was marginally insignificant (T = 1.89, p-value = 0.063). Given the estimate for the intercept is distant from the region of the data we chose to interpret this intercept as significant and report the full linear model's equation (Fig 2C).

The C:N ratios for $\delta^{13}C_{bulk}$ and $\delta^{15}N_{bulk}$ samples ranged from 3.4 to 32.5 (Fig 3), which was larger than for $\delta^2H_{bulk}$ samples. Muscle $\delta^{13}C_{bulk}$ appeared to increase with C:N ratio (Fig 3A), while muscle $\delta^{15}N_{bulk}$ appeared to decline with increasing the C:N ratio (Fig 3B). A simple linear model was not fit to these data because the relationships appeared more complex (Fig 3) than for $\delta^2H_{bulk}$ (Fig 2).

### Isotope ratios models

Regressing $\Delta\delta^2H$ against the percent hydrogen resulted in non-linear model having the lowest AIC score, but only one of its three terms (c) was significant (Fig 4A; Table 1). The linear relationship had a slightly higher AIC score (3 units), MSE, and MAE, but both terms were significant (Fig 4A; Table 1). Model efficiency was also higher for Eq 3, 43% of values were within 2 SDs, compared to 23% for Eq 1 (Table 1). The individual with the highest %H drove the differences between these two models (Fig 4A). Therefore, we selected a linear model as the top model for $\Delta\delta^2H$ (Eq 1; Table 1) and as a result there was no calculable maximum difference between muscle and lipid (i.e., no asymptote is present in a linear model). The $\Delta\delta^2H$ top model (Eq 1) estimated the precent hydrogen in a lipid-free sample would be 6.02% (Table 1; Fig 4A) which was not different from the measured value in the lipid-extracted samples (6.04%; $T_{20,117}$ = 0.06, p-value > 0.5).

Fitting of $\Delta\delta^{13}C$ provided strong support for non-linear models (Eqs 3 and 4; Fig 4B), which always had AIC values >40 units different from the linear model, lower MSE and MAE values (Table 1), and more homoscedastic residuals (S1 Fig). Model efficiency was also approximately twice as high in any non-linear model when compared to the linear model (~45% vs. 26%; Table 1). Eqs 3 and 4 had the lowest AIC scores, MSE, and MAE values, and all their

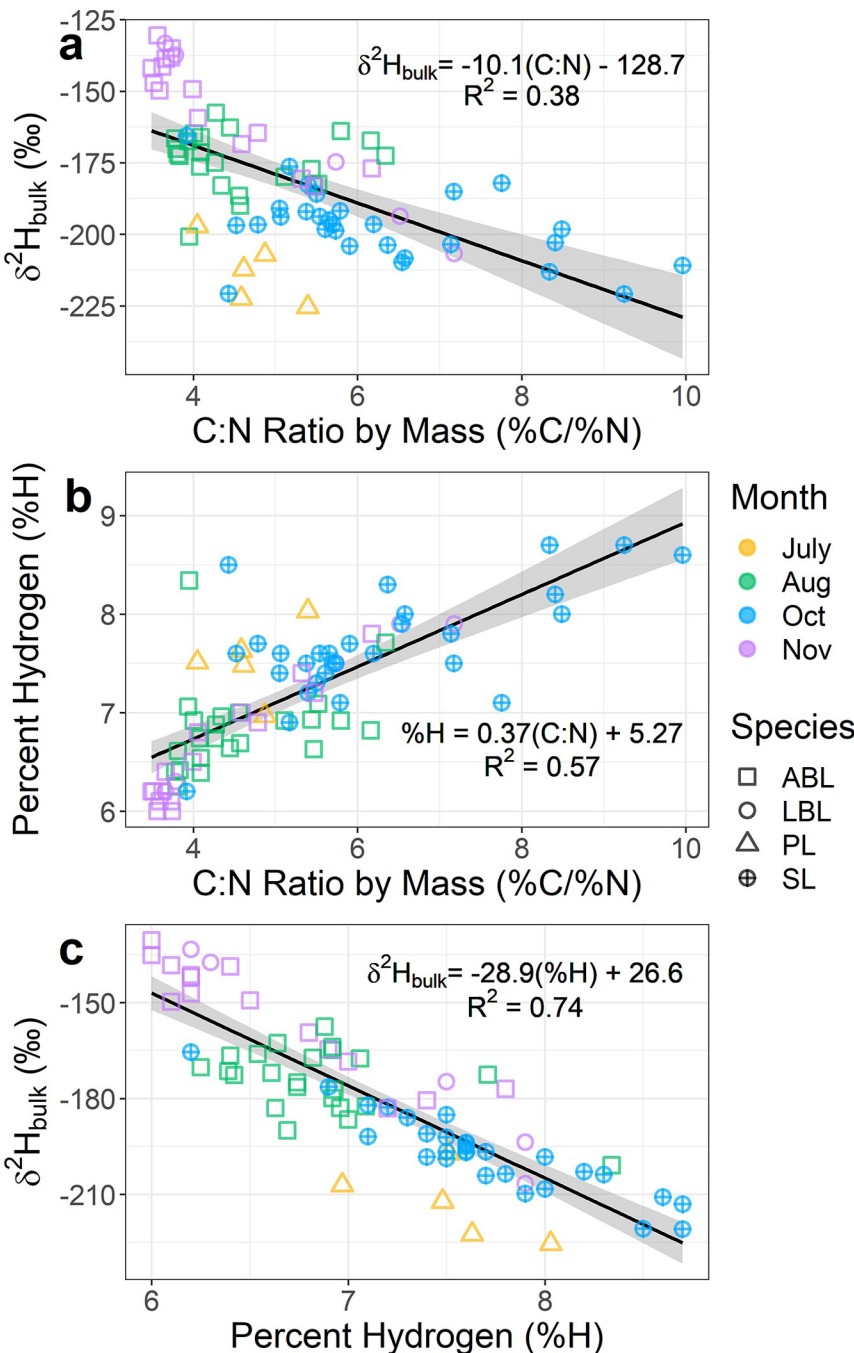

**Fig 2.** a) The $\delta^2H_{bulk}$ of muscle and the C/N ratio measured during $\delta^{13}C/\delta^{15}N$ analysis in a separate subsample. b) The percent hydrogen in a $\delta^2H_{bulk}$ sample regressed against the C/N ratio measured during $\delta^{13}C/\delta^{15}N$ analysis in another subsample. c) The muscle $\delta^2H_{bulk}$ sample regressed against the percent hydrogen which is measured simultaneously to $\delta^2H$. Also shown on plots are the linear fits (solid black lines), the 95% confidence intervals of the fit (grey region), the linear model's equation, and its $R^2$ values. Although not used for linear fitting, the samples are colored by month of collection, and the species collected is denoted by shape (ABL = American brook lamprey, LBL = least brook lamprey, SL = sea lamprey).

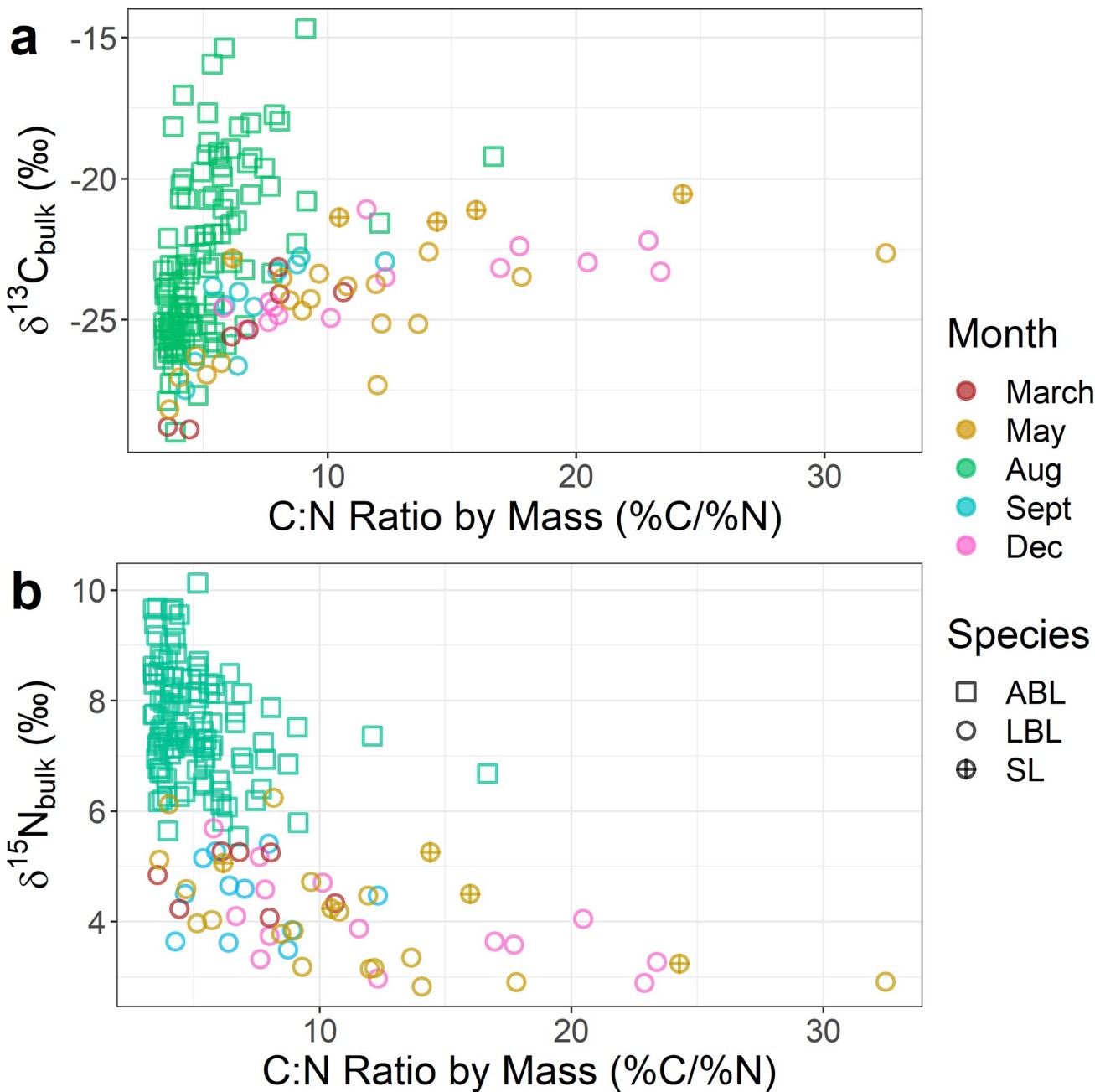

**Fig 3.** a) The $\delta^{13}C_{bulk}$ and b) the $\delta^{15}N_{bulk}$ of lamprey muscle and their C:N ratio, colored by the month in which they were collected. The species collected is denoted by shape (ABL = American brook lamprey, LBL = least brook lamprey, SL = sea lamprey).

terms were significant, but Eq 4 was a simpler model and was selected as the top model. The top model for $\Delta\delta^{13}C$ (Eq 4) estimated $\Delta_{ML}\delta^{13}C$ to be -6.15 ± 0.23‰ (mean ± SE, n = 119; Table 1) which was lower than the mean measured value (-5.03 ± 0.23‰, n = 57). The model also overestimated the C:N ratio (3.59 ± 0.001, mean ± SE) of a lipid-free sample ($T_{119,121}$ = 15.72, p-value < 0.0001); in measured samples it was 3.34 ± 0.02 (mean ± SE, n = 121). Visual inspection of the top model for $\delta^{13}C$ (Fig 4B) and its residuals (S1 Fig) showed that it was not entirely homoscedastic; residuals for points with a C:N of <3.5 tended to be positive.

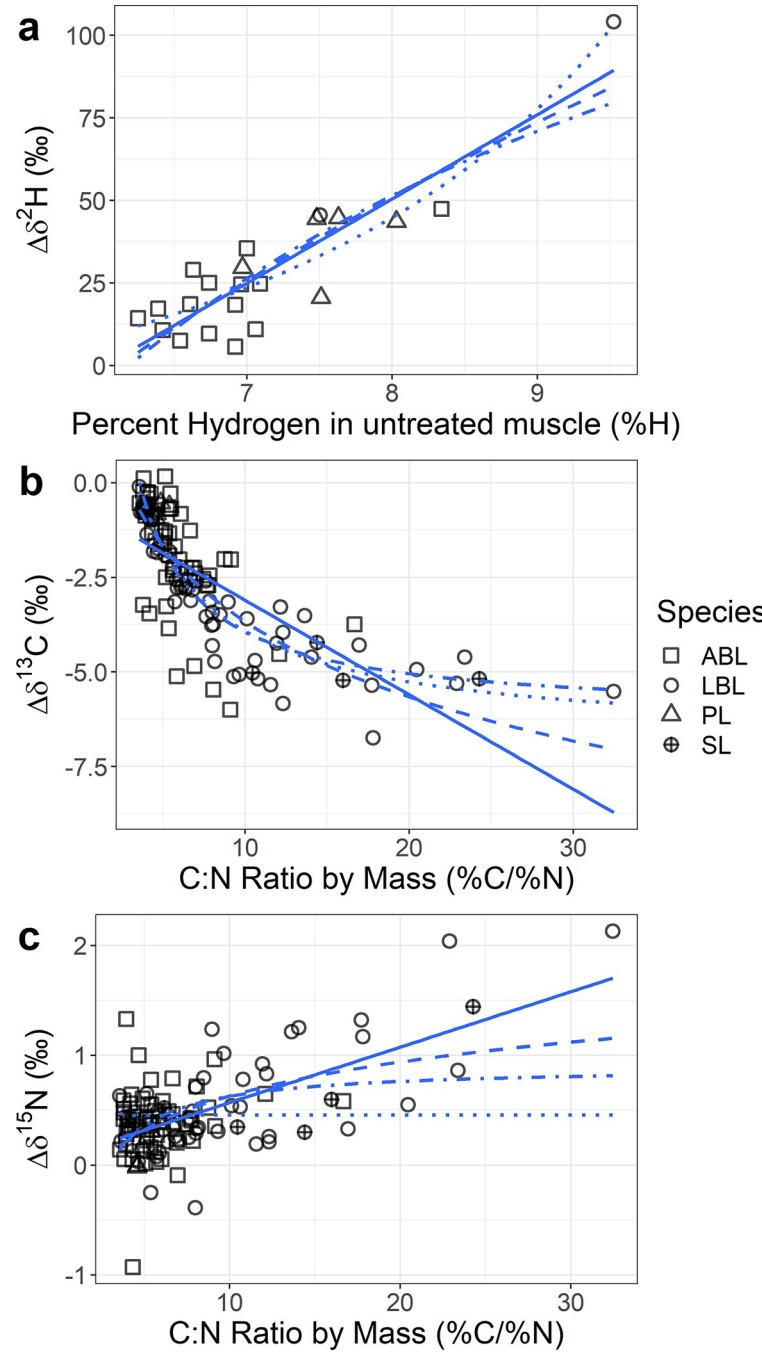

**Fig 4.** a) The $\Delta\delta^2$H ($\delta^2$H$_{extracted}$−$\delta^2$H$_{bulk}$) of muscle samples regressed against the percent hydrogen in the unextracted muscle sample. b) The $\Delta\delta^{13}$C ($\delta^{13}$C$_{extracted}$−$\delta^{13}$C$_{bulk}$) and the c) $\Delta\delta^{15}$N ($\delta^{15}$N$_{extracted}$−$\delta^{15}$N$_{bulk}$) of muscle samples regressed against the C:N ratio in the bulk muscle sample. The lines of best fit are overlain; the solid line is y = ax +b, the dashed line is y = a·ln(x) + b, the dotted line is y = (ax + b)/(x + c), and the dot-dash line is y = (ax + b)/x.

Lastly, $\Delta\delta^{15}$N was best fit by a linear model based on all measures (Table 1). The slope of $\Delta\delta^{15}$N in the linear model (Eq 1) was different from zero (0.050 ± 0.006‰, mean ± SE), but the intercept was not (Fig 4C; Table 1). Visually, as C:N approached seven, $\Delta\delta^{15}$N appeared to increase, suggesting for low C:N samples, lipids had only a small, or no, influence on $\Delta\delta^{15}$N (Fig 4C). The data were heavily right skewed, violating assumptions of normality in the

**Table 1. The model, equation number used in the present study, number of samples (n), model AIC value, the difference in AIC units (ΔAIC) from the top model in the family, mean squared error (MSE), mean absolute error (MAE), a measure of model efficiency, and parameter estimates (mean ± SE).**

| Isotope Ratio | Model | Eqn. # | N | AIC | ΔAIC | MSE | MAE | Model Efficiency | a | b | c |
|---|---|---|---|---|---|---|---|---|---|---|---|
| $\delta^2$H | $\Delta\delta^2$H ~ a · %H + b | 1 | 22 | 168 | 3 | 92.1 | 8.25 | 23% | 25.5 ± 2.9*** | -154 ± 21*** | NA |
| $\delta^2$H | $\Delta\delta^2$H ~ a · ln (%H) + b | 2 | 22 | 171 | 6 | 104.4 | 8.62 | 23% | -347 ± 47*** | -191 ± 24*** | NA |
| $\delta^2$H | $\Delta\delta^2$H ~ (a · %H + b) / (%H + c) | 3 | 22 | 165 | 0 | 72.9 | 7.09 | 41% | -76 ± 56 | 398 ± 346 | -12.7 ± 1.94*** |
| $\delta^2$H | $\Delta\delta^2$H ~ (a · %H + b) / (%H) | 4 | 22 | 174 | 9 | 119.1 | 9.00 | 14% | 227 ± 27*** | -1405 ± 191*** | NA |
| $\delta^{13}$C | $\Delta\delta^{13}$C ~ a · C:N + b | 1 | 121 | 385 | 57 | 1.35 | 0.92 | 26% | -0.250 ± 0.022*** | -0.607 ± 0.250*** | NA |
| $\delta^{13}$C | $\Delta\delta^{13}$C ~ a · ln (C:N) + b | 2 | 121 | 339 | 11 | 0.92 | 0.72 | 42% | 2.94 ± 0.36*** | -2.87 ± 0.18*** | NA |
| $\delta^{13}$C | $\Delta\delta^{13}$C ~ (a · C:N + b) / (C:N + c) | 3 | 121 | 328 | 0 | 0.82 | 0.66 | 46% | -6.79 ± 0.61*** | -23.17 ± 1.76*** | 1.41 ± 1.30*** |
| $\delta^{13}$C | $\Delta\delta^{13}$C ~ (a · C:N + b) / (C:N) | 4 | 121 | 328 | 0 | 0.84 | 0.66 | 44% | -6.15 ± 0.23*** | -22.05 ± 1.31*** | NA |
| $\delta^{15}$N | $\Delta\delta^{15}$N ~ a · C:N + b | 1 | 121 | 71 | 0 | 0.10 | 0.24 | 95% | 0.050 ± 0.0060*** | 0.064 ± 0.054 | NA |
| $\delta^{15}$N | $\Delta\delta^{15}$N ~ a · ln (C:N) + b | 2 | 121 | 87 | 16 | 0.11 | 0.25 | 93% | -0.40 ± 0.13*** | 0.45 ± 0.065*** | NA |
| $\delta^{15}$N | $\Delta\delta^{15}$N ~ (a · C:N + b) / (C:N + c) | 3 | 121 | 129 | 58 | 0.16 | 0.27 | 90% | 0.46 ± 0.04*** | -1.70 ± 0.22*** | -3.73 ± 0.37*** |
| $\delta^{15}$N | $\Delta\delta^{15}$N ~ (a · C:N + b) / (C:N) | 4 | 121 | 103 | 32 | 0.13 | 0.26 | 91% | 0.90 ± 0.091*** | -2.71 ± 0.52*** | NA |

Note: Model efficiency was calculated as the proportion of predicted minus observed values that were within twice the standard deviation of machine uncertainty (4‰ for $\delta^2$H, 0.4‰ for $\delta^{13}$C, 0.6‰ for $\delta^{15}$N).

*** denotes p-values < 0.001

models. The average $\Delta\delta^{15}$N for samples with C:N < 10 was 0.36‰ (SD ± 0.30, n = 98); the mean and SD for the $\Delta\delta^{15}$N for all data (n = 121) was ~0.1‰ higher (Fig 1).

## Seasonal difference between muscle and lipid

Visually there was no evidence that the measured difference between muscle and lipid $\delta^2$H ($\Delta_{ML}\delta^2$H) varied between months (Fig 5A and S2A Fig). An ANOVA confirmed this ($F_{3, 31}$ = 1.56, p > 0.2); the overall mean difference was 111‰ (SE = ± 21, n = 35; Fig 5A). In contrast, the $\Delta_{ML}\delta^{13}$C was different between months (ANOVA, $F_{3,53}$ = 5.05, p < 0.005; S2B Fig); March was different from all months, but no other month was different from another (Fig 5B). The mean $\Delta_{ML}\delta^{13}$C in March was -3.52‰ (SE = ± 0.67, n = 15) which was lower than other months (mean = -5.58‰, SE = ± 0.20, n = 42; Fig 5B). The variance in March was also larger (6.8‰) than other months (May = 2.9‰, Sept = 0.87‰, Dec = 0.46‰) and March included samples where the muscle and lipid had identical $\delta^{13}$C values (Fig 5B). The top model for $\Delta\delta^{13}$C (Eq 4) had an asymptote of -6.15 (Table 1) which was lower than the $\Delta_{ML}\delta^{13}$C measured value in March (-3.52 ± 0.67‰, n = 15) or all other months (-5.58 ± 0.20‰, n = 42).

## Discussion

The carbon scaffold of lipids in larval lampreys are often enriched in $^{13}$C relative to muscle, but the attached hydrogens are depleted in the heavy isotope ($^2$H; and therefore, enriched in the lighter isotope $^1$H). It is unclear why lipid synthesis should prefer a lighter isotope in one case (hydrogen), but not another (carbon). Larval lampreys are known to have unusual lipid origins, relying on muscle instead of the liver cells for synthesis [31]. Alternatively, larval lampreys may retain lipid scaffolding from other organisms they feed on, such as bacteria and algae [41], and then saturate any double bonds that occur primarily with $^1$H. Stream algae $\delta^{13}$C is variable and dependent on water velocity [42] and their lipids may be relatively enriched in $\delta^{13}$C compared to larval lamprey muscle.

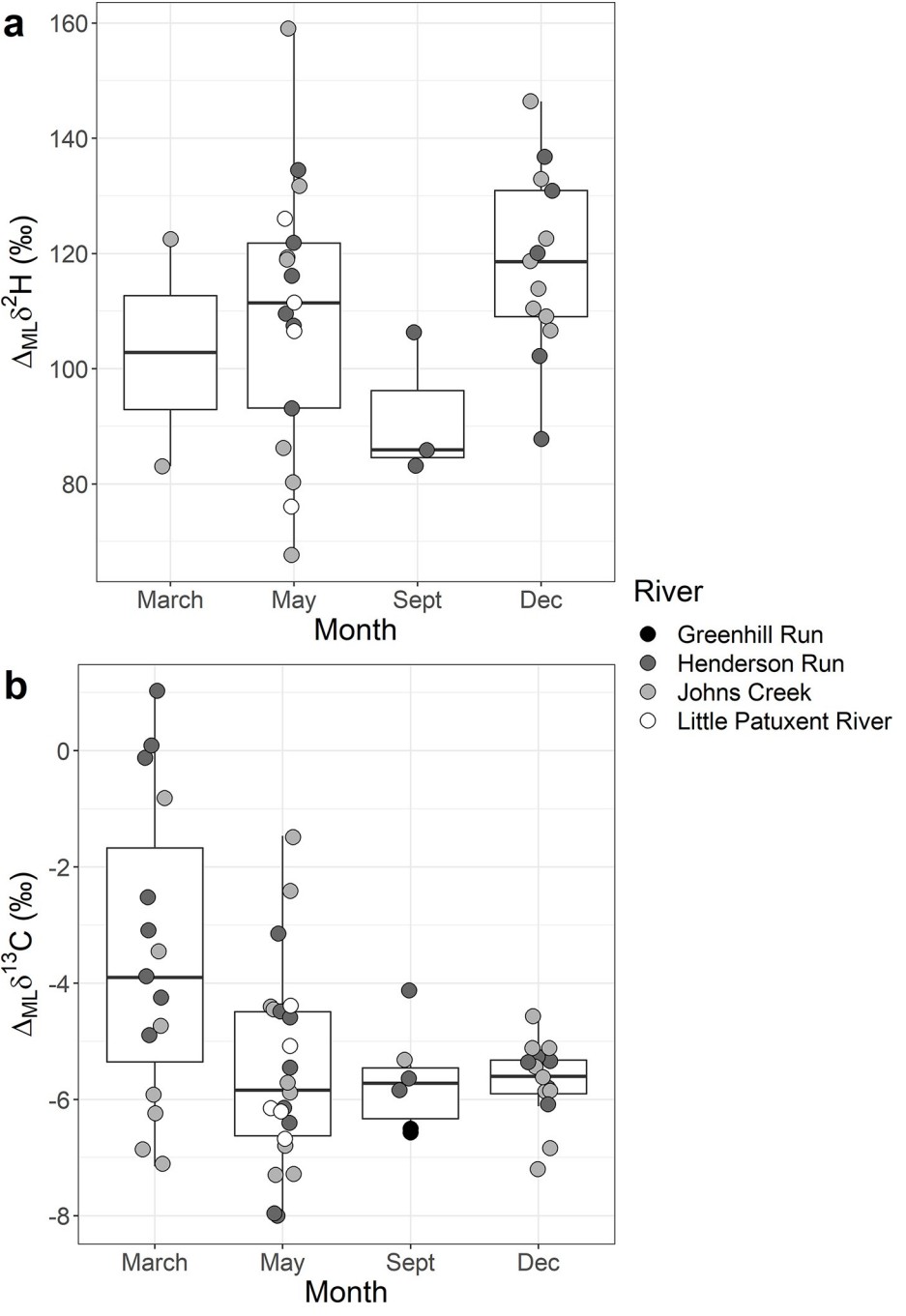

**Fig 5.** a) The $\Delta_{ML}\delta^2H$ ($\delta^2H_{extracted}-\delta^2H_{lipid}$) and b) the $\Delta_{ML}\delta^{13}C$ ($\delta^{13}C_{extracted}-\delta^{13}C_{lipid}$) plotted by the month of collection and color for the river they were collected from.

Larval lipid accumulation of $^{13}C$ may be related to their low metabolic rates; larvae sometimes require a decade or more to grow only a few grams [26]. Larval lampreys consume only a small fraction of lignin or cellulose in their diet (<10%) [43], ensuring much of the material they consume passes undigested through the gut; some ingested algae pass through larval lamprey guts and emerge alive [43, 44]. Presumably when feeding, larval lampreys quickly consume the few carbohydrates they encounter, and then rely upon energy from amino acids and

lipids for most metabolic costs. Only after quickly utilizing the relatively fast reacting $^{12}$C and excreting it as $^{12}CO_2$ or incorporating it into their own amino acids, would larvae begin to synthesize lipids. Since $^{12}$C is expelled or sequestered the body fatty acids may be relatively enriched in $^{13}$C.

The seasonal difference in $\delta^{13}$C between muscle and lipid ($\Delta_{ML}\delta^{13}$C; Fig 5B) suggests the enrichment in $^{13}$C occurs primarily after the spring algal bloom and may be related to a physiological shift in the larvae. Larval lamprey glycogen content in muscles, a more water-soluble energy store, varies seasonally, with levels rising in spring and then declining in summer before beginning to rise again in the fall, presumably in response to increase in the quality of detrital material during periods of higher diatom growth [31]. Glycogen can alter stable isotope signatures in organisms and also causes C:N values to rise [16]. Larval lampreys also deposit the greatest amount of lipids during periods of high diatom density [33]. Therefore, during the spring sampling (i.e., March), bulk muscle likely included the highest quantities of glycogen, and after lipid extraction the supernatant could include glycogen. Glycogen could have caused presumed lipids (liquid left after the supernatant evaporated) to be isotopically similar to lipid extracted muscle (S2 Fig). As the year progressed, these glycogen stores would have been reduced, and presumed lipids would better reflect pure lipid (S2 Fig). Larval lamprey glycogen and lipid content must be considered before interpreting their stable isotope ratios.

Trophic discrimination factors, and ultimately bulk muscle isotope values, can vary with diet [7]. Larval lamprey are detritivores, feeding on sources often high in carbon and hydrogen, but low in nitrogen for large portions of the year [45]. Larvae probably cannot afford to discriminate against nitrogen frequently (as a detritivorous fish [46]) but would need to routinely discriminate against carbon and hydrogen, which are always in excess. During periods of higher quality detrital intake (in most streams during algal blooms), larvae may behave more like omnivores, discriminating to varying or lesser degrees depending on nutritional needs and growth rates. To date, most stable isotope studies of fish have focused on carnivores, omnivores, and to a lesser extent herbivores [47]. Detritivores, especially those that grow slowly, are not understood as thoroughly, and caution is advised when relying on assumptions made from dissimilar species (especially as the dissimilarity increases).

We found lipid extraction increased $\delta^{15}$N (Fig 3C), and our value (~0.4‰) was close to the value (0.3‰) reported in a metanalysis of lipid extraction impacts on $\delta^{15}$N [48], although slightly lower (0.7‰) than found for fishes in another metanalysis [49]. Removal of lipids relies on solvents, and these solvents are likely to remove proteins that are lipophilic, which could alter $\delta^{15}$N [50]. Our work reinforces findings from prior work that researchers should measure $\delta^{15}$N prior to lipid extraction [48, 49]. However, as prior authors have cautioned, much of the present work on lipid extraction's effect on $\delta^{15}$N has focused on carnivorous and omnivorous fishes [49], and not detritivores. It may be anticipated that detritivores, which are likely less discriminatory during incorporation, have similar nitrogen isotopes throughout their bodies and may be less impacted by lipid extraction than traditional fish models. The increase we observed in lamprey $\delta^{15}$N as C:N ratios increased (Fig 3C), suggest that either: 1) $\delta^{15}$N is also manipulated in the body (albeit to a lesser extent than $\delta^2$H or $\delta^{13}$C) or 2) larval lampreys with higher lipid stores were feeding on isotopically different food sources than those with lower lipid stores.

The use of $\delta^2$H is an emerging field [36], and lipids are known to alter bulk isotope ratios [10, 12]. Although we did not measure lipid content of muscle directly (relying instead on a proxy: %H or C:N ratio), our work strongly supports literature cautions [10, 12] that lipids influence bulk $\delta^2$H measures (Fig 2). The C:N ratio has often been used to correct for lipids in samples [6, 11], but researchers measuring $\delta^2$H may not have access to C:N values. Even if C:N

ratios are available, $\delta^2$H and C:N ratios are often measured in a different subsample, as in the present study, because fewer isotope facilities can measure $\delta^2$H/$\delta^{13}$C/$\delta^{15}$N simultaneously and the cost often increases for such an analysis. As a result, the C:N ratio of the $\delta^2$H sample is often unknown, albeit it is likely similar to the value reported when measuring $\delta^{13}$C/$\delta^{15}$N because the samples were homogenized. However, it is challenging to ensure samples remain truly homogeneous given that melting point of lipids is near room temperature and lipids tend to separate from the protein component (T. M. Evans, personal observation); the increased variance seen as C:N ratios increased (Fig 2A) is likely a result of incomplete homogenization between samples. Additionally, $\delta^2$H samples are small ($<0.4$ mg), making them vulnerable to small changes in the percentage of lipid within a sample. We utilized the percentage of hydrogen in a sample (Fig 2B), which provided better fits than C:N values (Fig 3), and presumably better estimates of the effect of lipid on samples. As has been done for $\delta^{13}$C [6, 11, 14], to increase the efficacy of $\delta^2$H, a comprehensive examination across multiple groups correlating the percentage of hydrogen with the percentage of lipid and their effect on the isotope ratio is needed.

Although our work provides compelling evidence that larval lamprey lipids must be considered during stable isotope studies, we had two limitations, which need to be clarified by future research. First, we had relatively few samples to examine seasonal effects, especially for $\Delta_{ML}\delta^2$H (Fig 5). The seasonal influence on $\Delta_{ML}\delta^{13}$C is intriguing and suggests changes to the stable isotope pools inside the body. Second, we do not provide a mathematical correction for $\delta^2$H$_{bulk}$, $\delta^{13}$C$_{bulk}$, or $\delta^{15}$N$_{bulk}$ values based on a lipid proxy. While our work provides evidence that lipids are important to consider when measuring larval lamprey isotopes, there is evidence that the time of year (Fig 5) and site may be important to consider (Figs 2 and 3), presumably because they could influence the glycogen content in muscle. We recommend that researchers measuring larval lamprey stable isotope ratios, either run samples in duplicate or make *ad hoc* correction models for their systems. A more formal solution will require laboratory rearing of larvae under different conditions and more repeated measures in wild populations.

Our work suggests, that while lipids are often depleted in $^{13}$C relative to muscle [6], this is not a universal rule. However, larval lampreys follow expectations for hydrogen [10, 12], where lipids are depleted in the heavier isotope ($^2$H) relative to the muscle. Scientists working with isotopes in larval lampreys should anticipate lipid effects. To ensure they interpret their stable isotope ratios meaningfully, larval lamprey samples should be run in duplicate, once on bulk samples and once after lipid extraction. As differences between muscle and lipid appear to depend on the season, mathematical correction is more challenging. These findings should not be assumed to be true of juvenile and adult lamprey lipids. The physiological mechanism (s) to explain larval lamprey lipids is unknown but is necessary to fully explain this phenomenon. Such work will provide deeper insights into how isotopes in animal bodies can be manipulated.

## Supporting information

**S1 Fig. The residuals regressed against the fitted values of the $\Delta\delta^2$H (a-d), $\Delta\delta^{13}$C (e-h), $\Delta\delta^{15}$N (i-l) models, with the equation listed in each panel.**
(TIFF)

**S2 Fig.** The a) $\delta^2$H, and b) $\delta^{13}$C for lipid (grey) and lipid extracted muscle (white) samples for lamprey collected in Maryland. Boxplot midlines are the median, the boxes enclose the 1st-3rd quartiles (i.e., 50% of the data points), and the whiskers are 1.5 times this interquartile range. Measured values that boxplots are drawn from are represented as points and are jittered to

make them more easily distinguishable.
(TIFF)

**S1 File.**
(DOCX)

## Acknowledgments

We thank St. Mary's College of Maryland for providing support and sampling equipment and Dr. Jeffrey Lombardo and five anonymous reviewers for reading and making comments on earlier drafts.

## Author Contributions

**Conceptualization:** Thomas M. Evans.

**Data curation:** Thomas M. Evans.

**Formal analysis:** Thomas M. Evans.

**Investigation:** Thomas M. Evans.

**Methodology:** Thomas M. Evans, Shale Beharie.

**Resources:** Thomas M. Evans.

**Supervision:** Thomas M. Evans.

**Validation:** Thomas M. Evans.

**Visualization:** Thomas M. Evans.

**Writing – original draft:** Thomas M. Evans.

**Writing – review & editing:** Thomas M. Evans, Shale Beharie.

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
