## [Decision Letter · Decision Letter 0]

5 Jan 2023

PONE-D-22-27112Are lipids always light? Lipids in larval lampreys are high in 13C but low in 2H relative to musclePLOS ONE

Dear Dr. Evans,

Thank you for submitting your manuscript to PLOS ONE. After careful consideration, we feel that it has merit but does not fully meet PLOS ONE’s publication criteria as it currently stands. Therefore, we invite you to submit a revised version of the manuscript that addresses the points raised during the review process.

As you will see below, your manuscript has now received two reviews from experts in your field. Both the reviewers and myself agree that this manuscript has substantial value and would be of great interest to those working in stable isotope ecology. However, I also agree with both reviewers that the manuscript requires major revisions before it can be accepted for publication.

At times I found the manuscript difficult to follow. Specifically, I think the results need to be improved so that they are easier to understand and more directly address the original hypotheses. I thought the paper did a good job of setting up of the aims and goals, but that the methods and results did not clearly answer the hypotheses. For example, I think there needs to be more direct discussions in the results about the comparisons between treated and untreated samples, in particular for carbon. Those direct comparisons are not explicitly discussed in the results, despite it being stated as one of the main purposes of the paper. Given the introduction is clear about its hypotheses, addressing those hypotheses should be made equally clear in the results. The results regarding the difference in delta13C values between untreated and extracted samples isn’t mentioned in the results text and instead the results jump straight to focusing on the best fit model. I realize the comparisons between untreated and extracted can be determined from figure 3, but because they are not clearly discussed in the text, that important and baseline result gets lost to the reader.  Perhaps a simpler figure than figure three with direct comparisons between treated and untreated samples as well as some specific commentary on the differences between each group would make things clearer. Both reviewers have also raised valid concerns about the statistical analysis that need to be addressed or clearly rebutted. Again, some of these concerns may stem from a lack of clarity in the text.  

Please read over all the suggestions for improvements with particular attention to comments on improving the overall clarity, detail, methods, and flow of the manuscript. Once again, thank you for your interesting work.

We look forward to receiving your revised manuscript.

Kind regards,

Samantha E.M. Munroe

Academic Editor

PLOS ONE

Journal Requirements:

3. We note that you have stated that you will provide repository information for your data at acceptance. Should your manuscript be accepted for publication, we will hold it until you provide the relevant accession numbers or DOIs necessary to access your data. If you wish to make changes to your Data Availability statement, please describe these changes in your cover letter and we will update your Data Availability statement to reflect the information you provide

Reviewers' comments:

Reviewer's Responses to Questions

**Comments to the Author**

1. Is the manuscript technically sound, and do the data support the conclusions?

Reviewer #1: Yes

Reviewer #2: No

2. Has the statistical analysis been performed appropriately and rigorously? 

Reviewer #1: Yes

Reviewer #2: No

3. Have the authors made all data underlying the findings in their manuscript fully available?

Reviewer #1: Yes

Reviewer #2: Yes

4. Is the manuscript presented in an intelligible fashion and written in standard English?

Reviewer #1: Yes

Reviewer #2: Yes

5. Review Comments to the Author

Reviewer #1: I have had the opportunity to review the manuscript titled “Are lipids always light? Lipids in larval lampreys are high in 13C but low in 2H relative to muscle.” Overall, I found this to be an important study that is generally clear and well-written. The authors found that 13C in lampreys becomes depleted with lipid extraction instead of enriched, opposite the general pattern. This has important implications for ecological inferences with stable isotopes, as the authors point out it makes them appear more like predators instead of detritivores. As such, important caveats in isotope ecology such as these are important to address. I think the authors do a good job of highlighting the importance of this and in setting up and carrying out their study. I do have several issues that should be addressed before moving forward with the manuscript. Namely, other causal sources for the pattern the authors found should be included, such as difference in the effects of lipids and lipid extraction among trophic levels and potential effects of growth patterns throughout the season on isotopic discrimination. In addition to these perspectives on their results, the authors should include more information of the errors of their predictive models.

Title:

should the end of the title be “but low in 2H in muscle?” The authors used muscle as their main tissue, so it isn’t in relation to muscle, it is in muscle.

Abstract

Line 17: Change to ‘Isotope ratios…’

Introduction

Line 41: I wouldn’t call this manipulation internal sources because it is really discrimination of isotope values during incorporation.

Line 48: Nitrogen isotopes can be affected by lipid extraction in some tissues, which is why lipid extraction can be expensive because samples potentially need to be run twice, once lipid extracted for carbon and then another non-lipid extracted for nitrogen.

Line 57: comma before ‘and.’

Line 69: Over what sort of time period? The entire larval stage?

Line 72: Do the authors mean at the end of their larval stage or do they undergo a metamorphosis at the end of each summer that they are larvae. Please clarify.

Methods

Line 109: Suggest moving “with a high-powered setting” so it is after “immobilized” and before “when they were observed.” Delete “was activated.”

Table 1 and associated results. Use other forms of error MSE, etc… This should help in selecting for the correct model. Also deltaic should also be presented since that is what matters most.

Line 263: Awkward phrasing, maybe “fitted values closer to a CN of 3.5 had less variation than values further from 3.5.” Was there some pattern to the residuals that might be better fit by adding a variance structure as a random effect? See Zuur et al. 2009 Chp 4. Mixed effects models and extensions in ecology with R.

Line 204-207 and Table 1: Most lipid correction studies use standard measures of error between predicted and observed values and include mean squared error, mean absolute error, and proportion predicted values with < 0.5 per mil difference from observed values. Also, provide delta AIC and AIC weights. These are standard for these types of analyses and should be used here as a measure of error and to compare the models.

Discussion

Line 299: You mean low metabolic rates?

Consider trophic level in lipid effects. Other studies have found that lower trophic level species have less discrimination of carbon-13 during lipid synthesis, but I’m not aware of the effects for detritivores. See

Cloyed, CS, DaCosta, KP, Hodanbosi, MR, Carmichael RH (2020). The effects of lipid extraction on carbon-13 and nitrogen-15 values and use of lipid-correction models across tissues, taxa, and trophic groups. Methods in Ecology and Evolution 11, 751-762.

Also, consider how growth may affect the different patterns among months? Are there different growth patterns throughout the year, especially since the authors mention they are very slow growing. Oftentimes, more growth results in less discrimination. One potential explanation is that during growth, more available material is needed, and organisms are more likely to have less discrimination.

Lines 317: The Cloyed et al. 2020 above is a recent meta-analysis that also found lipid-extraction can affect nitrogen values. Can’t remember what their results were exactly but would be worth including here.

Line 344: Other work has also found similar patterns.

Patterson, HK, Carmichael, R.H. (2016). The effect of lipid extraction on carbon and nitrogen stable isotope ratios in oyster tissues: implications for glycogen-rich species. Rapid Communications in Mass Spectrometry 30, 2594-2600

Reviewer #2: Dear Authors, thanks for the opportunity to review this manuscript that explored the effect of lipids on isotopic signatures of carbon, nitrogen, and hydrogen in larvae lampreys, and the relationship with lipid proxies (%H and CN ratio). I am convinced that your manuscript deserves publication, but based on my experience and review; it may require some major revisions because this study has a number of flaws (kindly refer to the comments below and the comments in the manuscript pdf for more details), which makes it challenging to accept it for publication at this stage. My major comment is you need to clarify the objectives of the study, the statistical tests that you used, and the results. My concern is that the statistical analyses and results presented in the manuscript seem disconnected from the objective and hypotheses. The interpretation of some results observed in the results section is unclear in the discussion section and did not match with the hypotheses. Some results were presented in the result section but data were not presented in the manuscript neither in a table or a figure. The flow of arguments is not always easy to understand. I agree the importance of identifying the effect of lipids on carbon, hydrogen, and nitrogen isotopic signatures in larvae lampreys because such information will contribute to better interpreting the hydrogen isotopic values in lampreys and to better understanding the lipid effect on stable isotope signatures of carbon, nitrogen, and hydrogen. I’ve listed below some points that can be considered to continue improving the manuscript before publication:

Major comments

1- I would change the title because “light” is confusing. Did you mean depleted? Also, you measured nitrogen and it doesn’t appear in the title. An idea: “Are lipids always isotopically depleted? Comparison of carbon, nitrogen, and hydrogen isotopic values of lipids in larval lampreys relative to muscle”

Introduction

1- I found the paragraph at lines 49-61 confusing. It seems to have two ideas: one about the need to correct for the lipid effect for the carbon and hydrogen isotopes and not for nitrogen isotope, and the second one about the estimation of lipid content and the proxy associated (%H, CN ratio). I do not understand the idea of this paragraph. Did you talk about the extraction of lipids in sample that increase the cost of a study to measure stable isotope ratios? If yes, I would change the beginning of the sentence for something like this: “However, extraction of lipids in sample can increase the cost of the study and the number of samples necessary because carbon is analyzed on a lipid-extracted subsample while nitrogen is analyzed on a bulk subsample”. If you talk about the necessity to measure the lipid content, I think you must clarify this entire paragraph.

2- At lines 81-86, I think that the good approach is to account for the lipid effect either by extracting lipids or by correcting mathematically (if mathematical models were available and validated for lamprey) for carbon and hydrogen isotopes prior to trophic and foraging interpretation. It is well known in the literature that lipids may biases interpretation of food web structure and diet estimates. Indeed, it is necessary to correct for the presence of lipids where body condition vary substantially due to lipid accumulation or loss, especially in your case for larvae lampreys.

3- I found the objectives of the study very confusing at lines 87-98 and also the statistical analyses used. If I understand well, you had 2 objectives: 1) to evaluate C, N, H isotopic signatures of lipids and compare to isotopic signatures of larvae lamprey muscles lipid-free and bulk, and to see if there is difference between lipids and muscle in order to confirm or not the expectation that the δ2H will be enriched in muscle after lipid-extraction, the δ13C will be depleted, and the δ15N will be unchanged, and 2) to evaluate the best relationship (linear or not) between the difference ΔδxX in lamprey muscle vs. the lipid proxy (%H, CN)? I may be confused but you need to clarify the objectives of the study to avoid misleading.

4- Also, in an another manuscript, you will be able to establish a mathematical equation to correct for the lipid effect if samples were analyses with lipids (see Post et al. 2007, Logan et al. 2008, Lesage et al. 2010, GroB et al. 2021 and more other). It will be a great article.

5- You first hypothesis was that larvae lamprey lipids would be enriched in 13C relative to muscle, and therefore the δ13C of muscle would decline after lipid extraction. You had no data and results presented in our article about this hypothesis. You may presents a graph δ13Cbulk vs. δ13Clipid-free and test the relationship. The second hypothesis was that larvae lamprey lipids would be depleted in 2H. Same here, no data was presented in the article. Also, the larvae lamprey lipids would be depleted in 2H relative to muscle lipid-free? The third hypothesis was that the δ15N would be unchanged by lipid extraction. No data presented again. You may test it with a paired t-test and presents a graph δ15Nbulk vs. δ15Nlipid-free.

Methods

6- You used the passive form and the active form in this section. You may choose one or the other.

7- Lines 114-115, please add more information about the dissection. Which part did you dissect? Muscle? Location of the muscle sample? Dorsal? Lateral?

8- Lines 117-124, please add more information about the protocol you used. Did you dry using a freeze-drier? Did you freeze sample at -80 before? After drying, did you powdered your muscle sample with a mortar and pestle? Cryomill? How long did you wash your sample in the chloroform-methanol solution? In a ultrasonic bath? At 4°C? Ambient temperature?

9- Lines 144-145, you mentioned that you had the larvae lipid samples from Maryland only. What did you measure? Lipid content? Stable isotope ratios?

10- Lines 170-183, if you have not created these equations, you should put the references associated with equations 1 to 4 (e.g. Logan et al. 2008 for eqn 2, McConnaughey et McRoy for eqn 3…).

11- Lines 170-183, you need to clarify the utility of the four model used. Is it to establish the best relationship (linear or not) between the difference ΔδxY in lamprey muscle vs. the lipid proxy (%H, CN)? These models did not answer to your hypotheses previously mentioned.

12- You selected the best supported model with the AIC criteria but did you evaluate the models? Did you compare the predicted and observed ΔδxY values? You will obtain R2, MAE, and you will see if the slope is close to 1 (no difference between the predicted vs. the observed values)? I think it is what you did at lines 237-241 with the t-test but it was not clear enough.

13- Lines 199-201, I have not read in details about that, but if the larvae accumulate a lot of lipids in their bodies during spring and their conditions declined throughout the summer to undergoes a metamorphosis at the end of the summer which they do not feed and rely on accumulated lipid stores (lines 69-74), do you think that it could impact the lipid synthesis and metabolism? Do the lipid accumulation vs. the conversion of lipid for metabolism use the same procedure for synthesis? Also, at lines 207-209 and figure 4b, there is a difference among month. I think that we must account for the month effect in the analysis.

14- ΔMLδ2H and ΔMLδ13C values are only presented in figure 4 by months. I think you must presented these data in the article and in a table.

15- At lines 204-207, you evaluated the reliability of the top ΔxY model, I do not understand this analysis. Did you compare the isotopic signature of lipid-free of muscle vs. lipid sample? Please clarify.

16- At lines, 205-207, I also don't understand this analysis. Do you want to compare the predicted difference vs. the observed difference? The results must be presented in the manuscript. A table with all data?

Results

17- It is not usual to have the legend of figures in the main text (see at lines 219-226, 228-232, 243-248, 285-287).

18- In order to affirm that the relation is positive, negative, or “appear linear”, you must use statistical tests to support these affirmations (e.g. lines 215, 216-217, 224-226).

19- At lines 237-238, where the data is presented in the manuscript? Also, at lines 238-241, 257-261, and 265-266.

20- I think you need to present the relationship and equations (linear or not) on figures 1 and 2, and the significance of the relationship. If linear, you need to have a R2.

21- Line 275, you have only 2 samples for March and only 3 for September. I'm not sure that you can test it with an ANOVA with this small sample size for these 2 months.

Discussion

22- I did not review in details this section because we need to clarify the introduction, objectives, methods, and results before.

23- There are limitations in all studies, but none are mentioned in your study. I would have appreciated to have a section covering the potential limits or your study.

So thanks for this interesting manuscript that explored the lipid effect on stable isotope signatures in larvae lampreys. I think that your manuscript deserves publication following major revisions.

6. PLOS authors have the option to publish the peer review history of their article (what does this mean?). If published, this will include your full peer review and any attached files.

Reviewer #1: No

Reviewer #2: No

---

## [Author Response · Author response to Decision Letter 0]

8 Mar 2023

PONE-D-22-27112

Are lipids always light? Lipids in larval lampreys are high in 13C but low in 2H relative to muscle

PLOS ONE

Dear Dr. Evans,

Thank you for submitting your manuscript to PLOS ONE. After careful consideration, we feel that it has merit but does not fully meet PLOS ONE’s publication criteria as it currently stands. Therefore, we invite you to submit a revised version of the manuscript that addresses the points raised during the review process.

As you will see below, your manuscript has now received two reviews from experts in your field. Both the reviewers and myself agree that this manuscript has substantial value and would be of great interest to those working in stable isotope ecology. However, I also agree with both reviewers that the manuscript requires major revisions before it can be accepted for publication.

At times I found the manuscript difficult to follow. Specifically, I think the results need to be improved so that they are easier to understand and more directly address the original hypotheses. I thought the paper did a good job of setting up of the aims and goals, but that the methods and results did not clearly answer the hypotheses. For example, I think there needs to be more direct discussions in the results about the comparisons between treated and untreated samples, in particular for carbon. Those direct comparisons are not explicitly discussed in the results, despite it being stated as one of the main purposes of the paper. Given the introduction is clear about its hypotheses, addressing those hypotheses should be made equally clear in the results. 

The results regarding the difference in delta13C values between untreated and extracted samples isn’t mentioned in the results text and instead the results jump straight to focusing on the best fit model. I realize the comparisons between untreated and extracted can be determined from figure 3, but because they are not clearly discussed in the text, that important and baseline result gets lost to the reader. Perhaps a simpler figure than figure three with direct comparisons between treated and untreated samples as well as some specific commentary on the differences between each group would make things clearer. 

Both reviewers have also raised valid concerns about the statistical analysis that need to be addressed or clearly rebutted. Again, some of these concerns may stem from a lack of clarity in the text.

Please read over all the suggestions for improvements with particular attention to comments on improving the overall clarity, detail, methods, and flow of the manuscript. Once again, thank you for your interesting work.

We look forward to receiving your revised manuscript.

Kind regards,

Samantha E.M. Munroe

Academic Editor

PLOS ONE

 

Journal Requirements:

Following the need to include a marked and unmarked manuscript, we have chosen to include line numbers that match the marked document in this letter to allow reviewers to see exactly how their comments have been incorporated.

We apologize for issues around the style. We worked hard, but the links provided demonstrated that we failed with several requirements. We have carefully gone back through the document and tried to exactly match the manuscript to the examples.

We did not originally show residual plots but have now included these as a supplementary figure and cited them in the text. We felt this was the most appropriate way to address this concern, given most readers will be unlikely to want to see these, but, as both reviewers noted, we needed to do a better job of showing model fitting/testing.

All relevant data are now within the manuscript and its Supporting Information files.

 

Reviewers' comments:

Reviewer's Responses to Questions

Comments to the Author

1. Is the manuscript technically sound, and do the data support the conclusions?

Reviewer #1: Yes

Reviewer #2: No

2. Has the statistical analysis been performed appropriately and rigorously? 

Reviewer #1: Yes

Reviewer #2: No

3. Have the authors made all data underlying the findings in their manuscript fully available?

Reviewer #1: Yes

Reviewer #2: Yes

4. Is the manuscript presented in an intelligible fashion and written in standard English?

Reviewer #1: Yes

Reviewer #2: Yes

5. Review Comments to the Author

 

Reviewer #1: I have had the opportunity to review the manuscript titled “Are lipids always light? Lipids in larval lampreys are high in 13C but low in 2H relative to muscle.” Overall, I found this to be an important study that is generally clear and well-written. The authors found that 13C in lampreys becomes depleted with lipid extraction instead of enriched, opposite the general pattern. This has important implications for ecological inferences with stable isotopes, as the authors point out it makes them appear more like predators instead of detritivores. As such, important caveats in isotope ecology such as these are important to address. I think the authors do a good job of highlighting the importance of this and in setting up and carrying out their study. 

We thank the reviewer for their time and effort, their work has greatly improved the manuscript.

I do have several issues that should be addressed before moving forward with the manuscript. Namely, other causal sources for the pattern the authors found should be included, such as difference in the effects of lipids and lipid extraction among trophic levels and potential effects of growth patterns throughout the season on isotopic discrimination. In addition to these perspectives on their results, the authors should include more information of the errors of their predictive models.

The reviewer has made many excellent suggestions which we have worked to incorporate, and this has greatly improved the manuscript. We now more fully explore models fits and incorporated papers which they cited. We are thankful for their patience and interest in seeing the paper improved.

Title

Should the end of the title be “but low in 2H in muscle?” The authors used muscle as their main tissue, so it isn’t in relation to muscle, it is in muscle.

L4-6: The title has been reworked following both reviewers’ comments.

Abstract

Line 17: Change to ‘Isotope ratios…’

L30: Changed to mirror the preceding section dealing with carbon.

Introduction

Line 41: I wouldn’t call this manipulation internal sources because it is really discrimination of isotope values during incorporation.

L60-61: Changed ‘manipulation internal sources’ to ‘isotope discrimination during the generation of new biomolecules’ following the reviewer’s comments.

Line 48: Nitrogen isotopes can be affected by lipid extraction in some tissues, which is why lipid extraction can be expensive because samples potentially need to be run twice, once lipid extracted for carbon and then another non-lipid extracted for nitrogen.

L74-78: We have now added wording highlighting these points in the paragraph emphasizing the issues with nitrogen measurements when accounting for lipids.

Line 57: comma before ‘and.’

This section has been substantially rewritten.

Line 69: Over what sort of time period? The entire larval stage?

L99: Changed to clarify this refers to the entire larval period.

Line 72: Do the authors mean at the end of their larval stage or do they undergo a metamorphosis at the end of each summer that they are larvae. Please clarify.

L102-103: We have clarified metamorphosis occurs at the termination of the larval period.

Methods

Line 109: Suggest moving “with a high-powered setting” so it is after “immobilized” and before “when they were observed.” Delete “was activated.”

L145-146: Changed following the reviewer’s recommendation.

Line 204-207 and Table 1: Most lipid correction studies use standard measures of error between predicted and observed values and include mean squared error, mean absolute error, and proportion predicted values with < 0.5 per mil difference from observed values. Also, provide delta AIC and AIC weights. These are standard for these types of analyses and should be used here as a measure of error and to compare the models.

L371-377: These components have now been added to the manuscript and into Table 1. We have added a ‘model efficiency’ estimate, not using a set 0.5‰ value, but relying on two times machine standards. We felt this provided a fairer comparison given the variability in machine measures of each isotope ratio. Our approach to selecting models has also been updated in the results, though we continue to present all models to allow readers the opportunity to interpret these results as well.

Table 1 and associated results. Use other forms of error MSE, etc… This should help in selecting for the correct model. Also deltaic should also be presented since that is what matters most.

L352-403: Following the reviewer’s comment we have added ΔAIC, MSE, and MAE to Table 1 to help readers compare models. We have also used these to show how we selected the top models in the results.

Line 263: Awkward phrasing, maybe “fitted values closer to a CN of 3.5 had less variation than values further from 3.5.” Was there some pattern to the residuals that might be better fit by adding a variance structure as a random effect? See Zuur et al. 2009 Chp 4. Mixed effects models and extensions in ecology with R.

We considered mixed effects models, and explored using site as a random effect, but these models did not improve fits. The inclusion of MSE and the plots of the residuals will hopefully provide evidence that the models we consider include members that fit the data well. 

Discussion

Consider trophic level in lipid effects. Other studies have found that lower trophic level species have less discrimination of carbon-13 during lipid synthesis, but I’m not aware of the effects for detritivores. See: 

Cloyed, CS, DaCosta, KP, Hodanbosi, MR, Carmichael RH (2020). The effects of lipid extraction on carbon-13 and nitrogen-15 values and use of lipid-correction models across tissues, taxa, and trophic groups. Methods in Ecology and Evolution 11, 751-762.

L609-615: We have included this paper when we discuss glycogen in larval lampreys to further stress the importance of body stores and the crude nature of C:N values. We agree with the reviewer that glycogen is likely to play an important role in the isotope values we observed.

Also, consider how growth may affect the different patterns among months? Are there different growth patterns throughout the year, especially since the authors mention they are very slow growing. Oftentimes, more growth results in less discrimination. One potential explanation is that during growth, more available material is needed, and organisms are more likely to have less discrimination.

L582-607: We have added considerably to an older paragraph and an entirely new paragraph to the discussion to deal with these points (i.e., seasonality, growth rate, discrimination variation) and increased the number of sources discussing this.

Line 299: You mean low metabolic rates?

L594-596: We have deleted this sentence

Lines 317: The Cloyed et al. 2020 above is a recent meta-analysis that also found lipid-extraction can affect nitrogen values. Can’t remember what their results were exactly but would be worth including here.

L582-607: We have included this paper during this section, and stressed that prior work has often focused on fishes, but the importance of diet should also be considered.

Line 344: Other work has also found similar patterns.

Patterson, HK, Carmichael, R.H. (2016). The effect of lipid extraction on carbon and nitrogen stable isotope ratios in oyster tissues: implications for glycogen-rich species. Rapid Communications in Mass Spectrometry 30, 2594-2600

We agree with the reviewer that this work demonstrates that C:N values in an organism are a bulk measure and not necessarily reflective of lipids alone. The authors do a good job in this paper demonstrating these differences are driven by stored glycogen in tissues. However, when they measured tissue with high lipid content, as predicted, δ13C rose because the lipids were presumably depleted. We are not aware of a paper which has measured lipids directly and found that they are the driving component of rising δ13C values.

L582-594: Larval lamprey glycogen content has been studied before (O’Boyle and Beamish 1977), and these authors found that it changes seasonally. This was not included in our discussion, and should have been because of the importance of stressing that C:N is a bulk measure and does not necessarily reflect lipids. We have included Patterson and Carmichael now and discussed the possibility of glycogen’s impact on our stable isotope values.

O'Boyle, R. N., & Beamish, F. W. H. (1977). Growth and intermediary metabolism of larval and metamorphosing stages of the landlocked sea lamprey, Petromyzon marinus L. Environmental Biology of Fishes, 2(2), 103-120.

 

Reviewer #2: 

Dear Authors, thanks for the opportunity to review this manuscript that explored the effect of lipids on isotopic signatures of carbon, nitrogen, and hydrogen in larvae lampreys, and the relationship with lipid proxies (%H and CN ratio). I am convinced that your manuscript deserves publication, but based on my experience and review; it may require some major revisions because this study has a number of flaws (kindly refer to the comments below and the comments in the manuscript pdf for more details), which makes it challenging to accept it for publication at this stage. 

We thank the reviewer for their kind words, and their help in improving the manuscript.

The reviewer made numerous small edits to improve the grammar and readability of the manuscript. To keep this response concise, we have not noted when each has been accepted, but we have gone through and considered each one. However, whenever a comment was left, we have transcribed it here and responded to it. We thank the reviewer for the time they dedicated to this, we appreciate how much effort and care they took in reading the document.

My major comment is you need to clarify the objectives of the study, the statistical tests that you used, and the results. My concern is that the statistical analyses and results presented in the manuscript seem disconnected from the objective and hypotheses. The interpretation of some results observed in the results section is unclear in the discussion section and did not match with the hypotheses. Some results were presented in the result section but data were not presented in the manuscript neither in a table or a figure. The flow of arguments is not always easy to understand.

We have worked throughout the document to clarify why we approached the data in the way we did (including considerably rewriting the end of the introduction, L118-129). We have also added more explorations with linear models and hope that the review feels that it is easier to understand.

I agree the importance of identifying the effect of lipids on carbon, hydrogen, and nitrogen isotopic signatures in larvae lampreys because such information will contribute to better interpreting the hydrogen isotopic values in lampreys and to better understanding the lipid effect on stable isotope signatures of carbon, nitrogen, and hydrogen. I’ve listed below some points that can be considered to continue improving the manuscript before publication:

The reviewer worked very hard to provide a rich review. Their work has helped us as we improved the manuscript. We thank them for this effort and time commitment.

Major comments

1- I would change the title because “light” is confusing. Did you mean depleted? Also, you measured nitrogen and it doesn’t appear in the title. An idea: “Are lipids always isotopically depleted? Comparison of carbon, nitrogen, and hydrogen isotopic values of lipids in larval lampreys relative to muscle”

L4-6: Based on this comment and the first reviewer we have changed the title to better match the content of the paper and purpose of the work. We appreciate the reviewer providing a descriptive and concise title for consideration.

Abstract

L18-21: I would suggest to modify this sentence to clarify. Something like this: "... in Maryland. The isotopic ratios were measured in both bulk and lipid-free muscle samples, but also in extracted lipid samples."

L35-37: We have rewritten this sentence following the reviewer’s comment.

L27-28: I'm not sure that I would finish my abstract with this sentence. Why don't you change for the implication of your study?

L44-46: We have deleted the sentence and agree with the reviewer it was not appropriate. We feel the preceding sentence is a better summary of our findings and points to the wider impacts of our findings.

Introduction

L35: Please clarify. Enzymatic reaction? Metabolic reaction? Photosynthetic reaction?

L54: Clarified to enzymatic reaction in the text now following the reviewer’s comment.

L37: What do you mean by starting materials? food sources? Diet?

L56: We have clarified that this paper is dealing with dietary sources.

L38: What do you mean by predictable? Not sure that it is predictable. It is assumed 1‰ for carbon and 3‰ for nitrogen but it is highly variable among organisms, tissues, body condition, etc.

L57: We have deleted ‘predictable.’ Originally our intent was to imply that there are expectations based on enzymatic kinetics, but we agree it implied a discrete exact quantity, which is not true; the actual difference is quite variable.

L47: I would change the reference here. Because Logan et al. 2008 did not study that and its sentence said that most lipid classes contain no nitrogen had 3 references: Schmidt et al. 2003, Bodin et al. 2007, Post et al. 2007.

L66-68: We have removed this sentence.

L47-48: Have you a reference to support that affirmation?

L67-68: We have deleted this sentence.

1- I found the paragraph at lines 49-61 confusing. It seems to have two ideas: one about the need to correct for the lipid effect for the carbon and hydrogen isotopes and not for nitrogen isotope, and the second one about the estimation of lipid content and the proxy associated (%H, CN ratio). I do not understand the idea of this paragraph. Did you talk about the extraction of lipids in sample that increase the cost of a study to measure stable isotope ratios? If yes, I would change the beginning of the sentence for something like this: “However, extraction of lipids in sample can increase the cost of the study and the number of samples necessary because carbon is analyzed on a lipid-extracted subsample while nitrogen is analyzed on a bulk subsample”. If you talk about the necessity to measure the lipid content, I think you must clarify this entire paragraph.

L69-80: Reviewer 1 also recommended changes to this paragraph, and we have substantially altered it. This paragraph now deals with the issue of lipid extraction and mathematical corrections as preferred. The following paragraph then deals with C:N ratios and the use of this to mathematically predict the lipid-free value.

L50: I would add a sentence about the nitrogen here such as "In contrast, nitrogen isotopes are not affected by the presence of lipids..."

L75-76: Here we point to the mechanism that could alter N isotope ratios.

L77-78: High means enriched? And the diet of larvae lamprey is also highly variable and include detritus, terrestrial and aquatic sources? Is it a possible explanation? Why carbon ratios of lamprey larvae are unexplainably high? Do you have some possible explanations?

L107-110: We did not want to spend long periods of time covering the prior papers, so we kept the language here brief.

Prior to this work there was no good explanation for the enriched 13C values unless lipids drive them. The prior studies have considered the solutions the reviewer puts forward, and none have been satisfactory in fully explaining the behavior.

2- At lines 81-86, I think that the good approach is to account for the lipid effect either by extracting lipids or by correcting mathematically (if mathematical models were available and validated for lamprey) for carbon and hydrogen isotopes prior to trophic and foraging interpretation. It is well known in the literature that lipids may biases interpretation of food web structure and diet estimates. Indeed, it is necessary to correct for the presence of lipids where body condition vary substantially due to lipid accumulation or loss, especially in your case for larvae of lampreys.

This and the next comment are closely related We agree this is the best approach, but this has not been done for lamprey. 

3- I found the objectives of the study very confusing at lines 87-98 and also the statistical analyses used. If I understand well, you had 2 objectives: 1) to evaluate C, N, H isotopic signatures of lipids and compare to isotopic signatures of larvae lamprey muscles lipid-free and bulk, and to see if there is difference between lipids and muscle in order to confirm or not the expectation that the δ2H will be enriched in muscle after lipid-extraction, the δ13C will be depleted, and the δ15N will be unchanged, and 2) to evaluate the best relationship (linear or not) between the difference ΔδxX in lamprey muscle vs. the lipid proxy (%H, CN)? I may be confused but you need to clarify the objectives of the study to avoid misleading.

L124-129: We have worked to rewrite the last paragraph of the discussion and clearly introduce the objectives of this study, following the pattern laid out by the reviewer.

4- Also, in an another manuscript, you will be able to establish a mathematical equation to correct for the lipid effect if samples were analyses with lipids (see Post et al. 2007, Logan et al. 2008, Lesage et al. 2010, GroB et al. 2021 and more other). It will be a great article.

We appreciate the encouragement. We would like to do such work with both wild populations and rearing experiments.

5- You first hypothesis was that larvae lamprey lipids would be enriched in 13C relative to muscle, and therefore the δ13C of muscle would decline after lipid extraction. You had no data and results presented in our article about this hypothesis. You may present a graph δ13Cbulk vs. δ13Clipid-free and test the relationship. The second hypothesis was that larvae lamprey lipids would be depleted in 2H. Same here, no data was presented in the article. Also, the larvae lamprey lipids would be depleted in 2H relative to muscle lipid-free? The third hypothesis was that the δ15N would be unchanged by lipid extraction. No data presented again. You may test it with a paired t-test and presents a graph δ15Nbulk vs. δ15Nlipid-free.

L214-334: The results of these types of analyses are shown in Fig 4 because it is the ΔδxY (the difference between bulk and lipid-extracted samples). However, it is clear we did a poor job of stepping through this analysis and demonstrating the need for this figure. Therefore, we have added the recommendation to include three t-tests, examining if lipid extraction had any effect of samples a priori, and then exploring how a proxy for lipids might better predict this difference.

Methods

6- You used the passive form and the active form in this section. You may choose one or the other.

We have tried to correct these now.

L108-110: “…when they were observed outside the sediment, larvae were immobilized with a high-power setting.”

L145-146: This sentence was also highlighted by the first reviewer and has been rewritten.

7- Lines 114-115, please add more information about the dissection. Which part did you dissect? Muscle? Location of the muscle sample? Dorsal? Lateral?

L153-157: We have added detail of how muscle was dissected.

L116: I would put the permit number here in [ ]

L159: We have added the permit number in the manuscript now.

8- Lines 117-124, please add more information about the protocol you used. Did you dry using a freeze-drier? Did you freeze sample at -80 before? After drying, did you powdered your muscle sample with a mortar and pestle? Cryomill? How long did you wash your sample in the chloroform-methanol solution? In a ultrasonic bath? At 4°C? Ambient temperature?

L161-162, 169: We have added these details to the methods now.

L143-144: Add references here

L190: We have now cited Post et al 2007 here, as they use this language. 

9- Lines 144-145, you mentioned that you had the larvae lipid samples from Maryland only. What did you measure? Lipid content? Stable isotope ratios?

L190-191: This has been clarified, it was the stable isotope ratios.

L152-153: Were the samples powdered? Give more details about the weight in brackets (e.g. 0.350-0.500 mg).

The muscle samples were powder, but lipids were an oily fluid. We mention the appearance of lipids and muscles in the preparation section (L167-170). We have now added the amount of material packed in the description (L198-199).

L157: Which standards? I think that it is very important for H because of the air moisture (Vander Zander et al. 2016)

L204-205: We have added the H internal standards because these include ones available for purchase. The standards for 13C/15N were things such as powder deer hoof are not available outside of this lab and would not provide much help for comparison between labs.

10- Lines 170-183, if you have not created these equations, you should put the references associated with equations 1 to 4 (e.g. Logan et al. 2008 for eqn 2, McConnaughey et McRoy for eqn 3…).

Both equations 2 and 3 are cited as being drawn from Logan already. McConnaughey and McRoy did not suggest any of the equations we relied upon. We did not cite the linear model (because of its simplicity) or eqn 4 because it is a simplification of the Logan et al. model. We included the function in R for completeness that was used as well. For instance:

“We also fit the data with a non-linear model [11] (fit by the nls function in R)”

[11] Is the reference to Logan. We want to make it clear that we are relying on Logan’s excellent summary and not proposing a new approach. If the reviewer still feels we failed to attribute these we are open to specific language to alter the sentence.

11- Lines 170-183, you need to clarify the utility of the four models used. Is it to establish the best relationship (linear or not) between the difference ΔδxY in lamprey muscle vs. the lipid proxy (%H, CN)? These models did not answer to your hypotheses previously mentioned.

L214-225, 264-269: We have worked to clarify the introduction following the reviewer’s suggestions, added considerable explanation to this in the methods, and now feel this should more naturally explain why we used models at this point.

12- You selected the best supported model with the AIC criteria but did you evaluate the models? Did you compare the predicted and observed ΔδxY values? You will obtain R2, MAE, and you will see if the slope is close to 1 (no difference between the predicted vs. the observed values)? I think it is what you did at lines 237-241 with the t-test but it was not clear enough.

L246-248, 264-269: Based on this comment and reviewer 1’s comments, we have greatly increased the materials presented around the models and modified this section in the manuscript.

13- Lines 199-201, I have not read in details about that, but if the larvae accumulate a lot of lipids in their bodies during spring and their conditions declined throughout the summer to undergoes a metamorphosis at the end of the summer which they do not feed and rely on accumulated lipid stores (lines 69-74), do you think that it could impact the lipid synthesis and metabolism? Do the lipid accumulation vs. the conversion of lipid for metabolism use the same procedure for synthesis? Also, at lines 207-209 and figure 4b, there is a difference among month. I think that we must account for the month effect in the analysis.

L276-280: We assumed enzymatic processing during lipid synthesis would be the same year-round. Our testing around seasonality emerged after we began to explore the data. Initially we did not expect it to matter, but were surprised when the data began to suggest seasonality maybe important. Based on this and the first reviewer, we have added materials in the discussion related to glycogen production (L447-459), which does help explain seasonal differences.

L203-204: I don't understand this test. You compare the isotopic signature of lipid-free of muscle vs. lipid sample? Reformulate.

L204-207: I don't understand this analysis. You want to compare the predicted difference vs. the observed difference? The results must be presented in the article. A table with all data?

L264-269: We have modified this section to more clearly lay out how we are checking the model efficiency.

14- ΔMLδ2H and ΔMLδ13C values are only presented in figure 4 by months. I think you must present these data in the article and in a table.

Each point is individually plotted on these graphs (Fig 5), the data are also available as supplementary files (S2 Fig), but we feel that the figure provides a concise way to visualize these.

15- At lines 204-207, you evaluated the reliability of the top ΔxY model, I do not understand this analysis. Did you compare the isotopic signature of lipid-free of muscle vs. lipid sample? Please clarify.

L264-280: Both reviewers were uncertain about how we examined models. We modified the end of the methods to clarify.

16- At lines, 205-207, I also don't understand this analysis. Do you want to compare the predicted difference vs. the observed difference? The results must be presented in the manuscript. A table with all data?

We used the model to estimate a value that we also measured so we felt it appropriate to see how close the model was to the measured value. We have added another figure showing the measured lipid and the lipid extracted muscle (S2 Fig).

Results

L214-217: Did you test it? Not really scientific to mention "was not universal"? What is the meaning of universal?

L219-222: It is mentioned regressed against, but you have not the relationship on these figures and the equation of the linear regression? non-linear? Which test did you use?

L224-226: Which test did you use to affirm that? Give the name of the test and the equation.

18- In order to affirm that the relation is positive, negative, or “appear linear”, you must use statistical tests to support these affirmations (e.g. lines 215, 216-217, 224-226).

20- I think you need to present the relationship and equations (linear or not) on figures 1 and 2, and the significance of the relationship. If linear, you need to have a R2.

L298-334: We felt that our approach here relied too much on visual inspection of figures. Both reviewers clearly felt that our approach to interrogating our data was insufficient. As a result, we have now added simple linear models to the δ2H vs C:N, %H vs C:N, δ2H vs %H because these were very linear in appearance. For the δ13C/δ15N, we have modified the language here to make it clear this is a visual exploration, not a mathematical quantification, because the relationship looked more complex. 

L228: LE or not LE?

We have introduced the work ‘bulk’ to refer to dried muscle samples before lipid extraction. We added this as a subscript to the isotope ratio notation throughout the document.

L237-241: Where the data are presented?

Our statement refers to the linear model’s lack of asymptote. This was more a statement of type of model, and not a reference to the data directly. We have added information clarifying that these models can be found in Table 1 and located in Figure 4.

L243: I would put different color of each model to facilitate the visualization.

We currently use a different line type. We considered the inclusion of color, but found it made it harder to distinguish where models overlap because the layering would tend to imply the last drawn model was the top model.

L254: Which one? Try to be clearer. eqn 1 X for example

L382: Added a refence to the two non-linear equations that seemed to fit the data well.

L257-261: I did not see these data in the article.

L388-392: For this analysis we are using the model parameters (presented in table 1) to make estimates, which we present in the text. Therefore, all data are available to the reader.

17- It is not usual to have the legend of figures in the main text (see at lines 219-226, 228-232, 243-248, 285-287).

The PLOS style requires legends in text immediately after the paragraph the figures are presented in.

19- At lines 237-238, where the data is presented in the manuscript? Also, at lines 238-241, 257-261, and 265-266.

We are unsure what data are not presented in the manuscript. It appears the reviewer is concerned that by showing a difference between measured samples we are not presenting our lipid data (all other data are now presented directly in the text). We have added this as supplementary boxplot (S2 Fig) to ensure we have all materials in the text.

L274: You must test it with statistical test. Visually is not a good method.

We use the visual method first and then test this statement with an ANOVA.

21- Line 275, you have only 2 samples for March and only 3 for September. I'm not sure that you can test it with an ANOVA with this small sample size for these 2 months.

L508: We bring up this limitation now in the discussion. We agree that the sample size is very small for some months, and therefore we anticipate the power of the test to be lower. \\

L287: Figure was in black and white.

We apologize if the reviewer’s image was black and white. The uploaded figure should be in greyscale, and its shades of grey identify different rivers.

Discussion

22- I did not review in details this section because we need to clarify the introduction, objectives, methods, and results before.

We appreciate that the reviewer may have more comments after seeing the revised manuscript.

L290: Not sure I understand why you are talking about backbone. You analysis only muscle and lipid right?

L423: Here backbone refers to the carbon scaffolding. We have modified the language to try to highlight this and avoid confusion with the term as a definition for the bones in the back of vertebrates.

L299-307: This part of the discussion help to understand which results?

L432-433: This paragraph is still discussing the unusual accumulation of 13C in larval lipids. We have tried to clarify this in the topic sentence.

L326: You have only a proxy of the lipids and not the lipid content. I would change the formulation.

L442-459: We have modified the text to emphasize that we support lipid as important, but we are relying on a proxy not a direct measure.

L333-336: Did you powder your samples to insure homogenization?

We tried and have elaborated in the text, but as we note, the protein component does not mix with the lipid content and tend to separate. Since the sample sizes are so small, it is nearly impossible to ensure true homogenization for δ2H and to a lesser degree δ13C/δ15N samples.

23- There are limitations in all studies, but none are mentioned in your study. I would have appreciated to have a section covering the potential limits or your study.

L506-517: Following the reviewer’s comments, we have added a whole paragraph dealing with this topic now.

So thanks for this interesting manuscript that explored the lipid effect on stable isotope signatures in larvae lampreys. I think that your manuscript deserves publication following major revisions.

We thank the reviewer for all their work.

6. PLOS authors have the option to publish the peer review history of their article (what does this mean?). If published, this will include your full peer review and any attached files.

Do you want your identity to be public for this peer review? For information about this choice, including consent withdrawal, please see our Privacy Policy.

Reviewer #1: No

Reviewer #2: No

---

## [Decision Letter · Decision Letter 1]

20 Apr 2023

PONE-D-22-27112R1Are lipids always depleted? Comparison of hydrogen, carbon, and nitrogen isotopic values of lipids in larval lampreysPLOS ONE

Dear Dr. Evans,

Thank you for submitting your manuscript to PLOS ONE. After careful consideration, we feel that it has merit but does not fully meet PLOS ONE’s publication criteria as it currently stands. Therefore, we invite you to submit a revised version of the manuscript that addresses the points raised during the review process.

We look forward to receiving your revised manuscript.

Kind regards,

Dharmendra Kumar Meena

Academic Editor

PLOS ONE

Additional Editor Comments:

The article in its present form can not be accepted for publication and authors are advised to revise the article as major revision. The article particularly has flaws presentation in terms of coherence in result and discussion part. In introduction latest refences to be used at least 80 % beyond 2017 onwards.

Reviewers' comments:

Reviewer's Responses to Questions

**Comments to the Author**

1. If the authors have adequately addressed your comments raised in a previous round of review and you feel that this manuscript is now acceptable for publication, you may indicate that here to bypass the “Comments to the Author” section, enter your conflict of interest statement in the “Confidential to Editor” section, and submit your "Accept" recommendation.

Reviewer #2: All comments have been addressed

Reviewer #3: All comments have been addressed

2. Is the manuscript technically sound, and do the data support the conclusions?

Reviewer #2: Yes

Reviewer #3: Yes

3. Has the statistical analysis been performed appropriately and rigorously? 

Reviewer #2: Yes

Reviewer #3: Yes

4. Have the authors made all data underlying the findings in their manuscript fully available?

Reviewer #2: Yes

Reviewer #3: Yes

5. Is the manuscript presented in an intelligible fashion and written in standard English?

Reviewer #2: Yes

Reviewer #3: Yes

6. Review Comments to the Author

Reviewer #2: Dear Editor and Authors, thanks for the opportunity to review this updated manuscript “Are lipids always depleted? Comparison of hydrogen, carbon, and nitrogen isotopic values of lipids in larval lampreys” that explored the effect of lipids on isotopic signatures of carbon, nitrogen, and hydrogen in larvae lampreys, the relationship of the difference between muscle samples before and after lipid extraction, and lipid proxies (%H and CN ratio), and the difference of isotopic signatures between muscle and lipid samples. I can testify that the authors have greatly improved the manuscript based on my previous comments and those made by the other reviewers. I’ve listed below some minor points that can be considered to continue improving the manuscript before publication.

General comments

1- There are few typos’ errors throughout the manuscript, I suggest that the authors reread the manuscript and pay attention to it. For example, here are some of them:

a. Line 27, a comma is missing between nitrogen, and hydrogen

b. Line 41, a comma is missing after natural systems

c. Line 54, a comma is missing after hydrogen (2H)

d. Line 201, a space is missing between (1) and where

e. Line 204, a space is missing between (2) and and

f. Line 293, a space is missing before while

g. Line 384, a space is missing between δ13C and between

h. Line 250, paragraph title is not written with the same writing style

i. etc…

2- I find that the context for the first hypothesis about the δ13C depletion after lipid extraction at line 106 is confusing. We did not understand why it could have a δ13C depletion after lipid extraction. According to the literature, if you removed deleted lipids in organism (or mathematically correct for them), the organism will become enriched in δ13C values. Also, at line 53, the authors mentioned that lipids are depleted in δ13C and δ2H. If depleted lipids are removed in animal tissues, its tissues will be enriched in δ13C, δ2H, and possibly in δ15N due to solvent that removed protein that are lipophilic. At lines 88-91, the authors mentioned that the δ13C values of larvae lamprey become unexplainably enriched after mathematical corrections, thus lipids seem depleted in δ13C for this specific example and it is consistent with the literature. In the literature, there is a negative relationship between CN and δ13C values, as the CN values increase, the δ13C values in tissues decreases. However, for larvae lampreys, there is a positive relationship between CN and δ13C values (line 72), which is very intriguing. When the CN is high, the δ13C is also high. Did the authors expect a depletion in δ13C values of muscle after lipid extraction due to this relationship? Why did the authors expect a δ13C depletion after lipid extraction? I would suggest clarifying the idea here because it is not intuitive.

Minor comments

Abstract:

1- Line 20, I would change “source contributions” for “dietary source contributions”.

2- Line 22, I would remove “for”.

3- Line 26, I would add a sentence about nitrogen isotope in order to introduce the objective that include the three isotopes.

4- Line 28, I would change “novel” for “different”.

5- Lines 31-32, I would change “bulk and lipid-extracted muscles, as well as the extracted lipids” for “bulk and lipid-extracted muscle samples, as well as in the extracted lipid samples”.

6- Line 36, I would replace “but” by “and” and what is the meaning of negative? The δ13C of lipids is enriched compared to the δ13C values of muscle? Please clarify and modify the wording.

Introduction:

7- Line 56, I would suggest adding something about nitrogen. This point is not mentioned anywhere in the article.

8- Lines 59-60, I would suggest changing “by chemically extracting them from a sample or mathematically correcting for them prior to interpretation” for “by chemically extracting them from a sample before stable isotope analysis or mathematically correcting for them a posteriori using validated normalization model” to clarify.

9- Lines 61-62, I would change “the number of samples” for “the number and quantity of samples as lipid extraction can remove lipophilic proteins which could alter nitrogen stable isotope ratios.”.

10- Lines 62-65, please clarify these sentences regarding the samples used for the carbon vs. nitrogen analysis. You may change for “The best practice is to run the lipid-extracted sample for carbon and hydrogen analysis and the bulk sample for nitrogen analysis. If workers choose to account for lipids a posteriori through a mathematically correction, they should use a validated model that typically used a proxy of lipid content and isotopic values of the bulk samples.”

11- Lines 66-72, I would suggest adding a few sentences about the relationship (or not) between CN ratio and δ2H, but also %H and δ2H. This will add context to the objectives/hypotheses at lines 107-108.

12- Line 72, could the lipids of larvae lampreys have a role to play in this relationship? Are there any possible explanations/hypotheses?

13- Line 90, please clarify what it is marine predators and why the δ13C signatures of larvae lampreys resemble more to them? Because at line 96, when lipids are extracted (and it correct to do that), they are entirely dependent on allochthonous detritus.

14- At line 106, the authors hypothesized that δ13C would be depleted in muscle after lipid extraction but according to the literature, if deleted lipids in organism are removed, the organism muscles will become enriched. Something is missing to better understand why the authors hypothesized that. Because at lines 88-91, δ13C values of larvae remain unexplainably enriched after mathematically corrections. I would suggest clarifying the idea in the introduction.

Methods:

15- Line 142, did the authors measure the % of lipid in the supernatant?

16- Line 169, please confirm that the exact values are 0.2 to 1.2mg? Usually, it is 1.0 to 1.2mg or around 0.500mg. The range is very large.

17- Line 227, do you think it can have a year effect on stable isotope signatures? It could be confused with the month.

Results:

18- Lines 264-267, I would remove these sentences because it is not a result. I would be more concise in the result section. It is not necessary to repeat the why of the analysis (e.g., lines 265-267, 269-272).

19- Lines 269-275, results are from linear regression? I would also write the r value in brackets along with the F and p values).

Discussion:

20- Lines 456-458, please clarify the link between running analysis in duplicate and the seasonal effect on mathematical correction? It seems 2 ideas from me. It is not the mathematical correction that depend on the season but the difference muscle-lipid that depend on the season? We run analysis in duplicate because δ13C is run on lipid-extracted sample while δ15N is run on bulk sample because lipid extraction affects δ15N values.

Figure 1. It would be interesting to add boxplot of the lipid isotopic values beside the isotopic values of lamprey muscles (bulk and lipid-extracted) for the three isotopes. It would facilitate the visualization of the isotopic values of lamprey’s lipids.

Figure 2a. Please write the whole equation, it is truncated.

So, thanks for this interesting manuscript that explored the lipid effect on stable isotope signatures in larvae lampreys. It is very intriguing why the lamprey’s lipids are enriched in δ13C instead to be depleted and the positive relationship between CN and δ13C values.

Reviewer #3: Dear authors,

This interesting piece of work presents a study on the stable isotope ratios of carbon, nitrogen, and hydrogen in the muscle of four lamprey species. The objective of the study was to determine if the stable isotope ratios of these elements behaved as expected, or if the lipids in the muscles presented novel isotopic behavior. I feel the author deserves publication of this work, however there are few queries that need to be addressed first.

Abstract

Overall, the abstract presents a clear and concise summary of the study's results and provides sufficient detail for the reader to understand the methodology and findings.

Introduction

It is well-written and informative. It provides a clear explanation of stable isotope ratios and their importance in studying biological processes. It also addresses the issue of accounting for lipids in stable isotope analysis, which is an important consideration that researchers need to take into account when interpreting results. The mention of the specific example of larval lampreys and the discrepancies in their stable isotope ratios adds an interesting layer to the discussion. However, I feel that few more recent works are available which can be incorporated.

Discussion

Perfectly fine, and issues detected.

My general view is the paper can be accepted with slight changes as suggested.

7. PLOS authors have the option to publish the peer review history of their article (what does this mean?). If published, this will include your full peer review and any attached files.

Reviewer #2: No

Reviewer #3: No

---

## [Author Response · Author response to Decision Letter 1]

15 May 2023

Reviewer #2: Dear Editor and Authors, thanks for the opportunity to review this updated manuscript “Are lipids always depleted? Comparison of hydrogen, carbon, and nitrogen isotopic values of lipids in larval lampreys” that explored the effect of lipids on isotopic signatures of carbon, nitrogen, and hydrogen in larvae lampreys, the relationship of the difference between muscle samples before and after lipid extraction, and lipid proxies (%H and CN ratio), and the difference of isotopic signatures between muscle and lipid samples. I can testify that the authors have greatly improved the manuscript based on my previous comments and those made by the other reviewers. I’ve listed below some minor points that can be considered to continue improving the manuscript before publication.

General comments

1- There are few typos’ errors throughout the manuscript, I suggest that the authors reread the manuscript and pay attention to it. For example, here are some of them:

a. Line 27, a comma is missing between nitrogen, and hydrogen

b. Line 41, a comma is missing after natural systems

c. Line 54, a comma is missing after hydrogen (2H)

d. Line 201, a space is missing between (1) and where

e. Line 204, a space is missing between (2) and and

f. Line 293, a space is missing before while

g. Line 384, a space is missing between δ13C and between

h. Line 250, paragraph title is not written with the same writing style

i. etc…

We have gone through the document and made these corrections. We have also gone through the document again to correct some instances where spaces and minor grammatical issues were missed in the last revision.

2- I find that the context for the first hypothesis about the δ13C depletion after lipid extraction at line 106 is confusing. We did not understand why it could have a δ13C depletion after lipid extraction. According to the literature, if you removed depleted lipids in organism (or mathematically correct for them), the organism will become enriched in δ13C values. Also, at line 53, the authors mentioned that lipids are depleted in δ13C and δ2H. If depleted lipids are removed in animal tissues, its tissues will be enriched in δ13C, δ2H, and possibly in δ15N due to solvent that removed protein that are lipophilic. At lines 88-91, the authors mentioned that the δ13C values of larvae lamprey become unexplainably enriched after mathematical corrections, thus lipids seem depleted in δ13C for this specific example and it is consistent with the literature. In the literature, there is a negative relationship between CN and δ13C values, as the CN values increase, the δ13C values in tissues decreases. However, for larvae lampreys, there is a positive relationship between CN and δ13C values (line 72), which is very intriguing. When the CN is high, the δ13C is also high. Did the authors expect a depletion in δ13C values of muscle after lipid extraction due to this relationship? Why did the authors expect a δ13C depletion after lipid extraction? I would suggest clarifying the idea here because it is not intuitive.

We have worked to clarify the logic behind the hypotheses. We summarize our thinking here to clarify as well. If tissues of larval lamprey δ13C are corrected with standard equations, they will rise because all current equations anticipate that lipids will have lower δ13C values than muscle. This results in values that are so high that the lampreys can only be explained by sources outside the system; even piscivorous fishes in the same stream have lower δ13C. Another possibility, the one we support, suggests that something is wrong with the expectations around lipids.

Minor comments

Abstract:

1- Line 20, I would change “source contributions” for “dietary source contributions”.

We have made the change following the reviewer’s comment.

2- Line 22, I would remove “for”. 

We retained ‘for’ as its removal causes the sentence to be grammatically incorrect (‘for’ is part of a multi-word verb starting with ‘accounted’).

3- Line 26, I would add a sentence about nitrogen isotope in order to introduce the objective that include the three isotopes.

We have added a clause to the sentence to make it appear more naturally in the abstract.

4- Line 28, I would change “novel” for “different”.

We have made the change following the reviewer’s comment.

5- Lines 31-32, I would change “bulk and lipid-extracted muscles, as well as the extracted lipids” for “bulk and lipid-extracted muscle samples, as well as in the extracted lipid samples”.

We have made the change following the reviewer’s comment.

6- Line 36, I would replace “but” by “and” and what is the meaning of negative? The δ13C of lipids is enriched compared to the δ13C values of muscle? Please clarify and modify the wording.

We have modified the sentence. We moved the word negative closer to the contrast in the statement (le muscle – lipid) to make it clearer.

Introduction:

7- Line 56, I would suggest adding something about nitrogen. This point is not mentioned anywhere in the article.

We have clarified the origin for discrimination during lipid synthesis and added a sentence at the end of the paragraph expressly dealing with nitrogen.

8- Lines 59-60, I would suggest changing “by chemically extracting them from a sample or mathematically correcting for them prior to interpretation” for “by chemically extracting them from a sample before stable isotope analysis or mathematically correcting for them a posteriori using validated normalization model” to clarify.

We have made the change following the reviewer’s comment.

9- Lines 61-62, I would change “the number of samples” for “the number and quantity of samples as lipid extraction can remove lipophilic proteins which could alter nitrogen stable isotope ratios.”.

We have not added ‘and quantity’ because it is not clear how number and quantity are different, but we have added a period a source following the reviewer’s comment.

10- Lines 62-65, please clarify these sentences regarding the samples used for the carbon vs. nitrogen analysis. You may change for “The best practice is to run the lipid-extracted sample for carbon and hydrogen analysis and the bulk sample for nitrogen analysis. If workers choose to account for lipids a posteriori through a mathematically correction, they should use a validated model that typically used a proxy of lipid content and isotopic values of the bulk samples.”

We have modified the sentence following the reviewer’s comment.

11- Lines 66-72, I would suggest adding a few sentences about the relationship (or not) between CN ratio and δ2H, but also %H and δ2H. This will add context to the objectives/hypotheses at lines 107-108.

Relatively little has been done with δ2H in larval lampreys, but we cite them here and note the dissertation where bulk samples were measured for δ2H and C/N analysis.

12- Line 72, could the lipids of larvae lampreys have a role to play in this relationship? Are there any possible explanations/hypotheses?

We have now explicitly stated that these correlations suggest lipids alter bulk values in stable isotopes.

13- Line 90, please clarify what it is marine predators and why the δ13C signatures of larvae lampreys resemble more to them? Because at line 96, when lipids are extracted (and it correct to do that), they are entirely dependent on allochthonous detritus.

We have deleted this cause because, while true, it is a qualitative observation. The point we were trying to make is that larval lamprey δ13C can be the highest value in streams. It is already clear from the introduction that larval lamprey δ13C is unusual.

14- At line 106, the authors hypothesized that δ13C would be depleted in muscle after lipid extraction but according to the literature, if deleted lipids in organism are removed, the organism muscles will become enriched. Something is missing to better understand why the authors hypothesized that. Because at lines 88-91, δ13C values of larvae remain unexplainably enriched after mathematically corrections. I would suggest clarifying the idea in the introduction.

We have clarified this is based on lamprey work. We expect that δ13C will decline in muscle after lipid extraction based on larval lamprey reports, not on the general trend expected for other species. We do not assume larval lamprey will have the same response as other groups because prior literature has already observed unusual behavior in their isotope ratios.

Methods:

15- Line 142, did the authors measure the % of lipid in the supernatant?

We did not measure the % lipid in the supernatant. 

16- Line 169, please confirm that the exact values are 0.2 to 1.2mg? Usually, it is 1.0 to 1.2mg or around 0.500mg. The range is very large.

This is the correct range, generally we tried to pack around 0.5 mg, but for lipids especially we preferred to use smaller quantities.

17- Line 227, do you think it can have a year effect on stable isotope signatures? It could be confused with the month.

We did not measure across multiple years but given larval dependence on benthic sources we would not expect isotopic signatures to vary greatly year to year.

Results:

18- Lines 264-267, I would remove these sentences because it is not a result. I would be more concise in the result section. It is not necessary to repeat the why of the analysis (e.g., lines 265-267, 269-272).

We deleted the rationale for the approach, leaving these in entirely in the methods now.

19- Lines 269-275, results are from linear regression? I would also write the r value in brackets along with the F and p values).

We have included the Pearson’s correlation coefficient (ρ).

Discussion:

20- Lines 456-458, please clarify the link between running analysis in duplicate and the seasonal effect on mathematical correction? It seems 2 ideas from me. It is not the mathematical correction that depend on the season but the difference muscle-lipid that depend on the season? We run analysis in duplicate because δ13C is run on lipid-extracted sample while δ15N is run on bulk sample because lipid extraction affects δ15N values.

We have changed the sentence to explain more clearly what we mean, as written, the reviewer is correct, it sounds as if we are asking people to run four analyses for each sample.

Figure 1. It would be interesting to add boxplot of the lipid isotopic values beside the isotopic values of lamprey muscles (bulk and lipid-extracted) for the three isotopes. It would facilitate the visualization of the isotopic values of lamprey’s lipids.

We considered this and went back and forth on how best to present these data. Ultimately, we left the figure as is for three reasons: 1) the hypothesis being evaluated only considers the difference before and after lipid extraction (i.e., there is no test associated with the lipids to reference when these data would be introduced), 2) the δ15N will not have a lipid value, 3) those lipid data are already available in S2 Fig.

Figure 2a. Please write the whole equation, it is truncated.

We have moved the equations to be fully in frame, we apologize that we missed this.

So, thanks for this interesting manuscript that explored the lipid effect on stable isotope signatures in larvae lampreys. It is very intriguing why the lamprey’s lipids are enriched in δ13C instead to be depleted and the positive relationship between CN and δ13C values.

We thank the reviewer for their comments and their work throughout this process. Their efforts have refined and improved the manuscript at all levels.

Reviewer #3: Dear authors,

This interesting piece of work presents a study on the stable isotope ratios of carbon, nitrogen, and hydrogen in the muscle of four lamprey species. The objective of the study was to determine if the stable isotope ratios of these elements behaved as expected, or if the lipids in the muscles presented novel isotopic behavior. I feel the author deserves publication of this work, however there are few queries that need to be addressed first.

Abstract

Overall, the abstract presents a clear and concise summary of the study's results and provides sufficient detail for the reader to understand the methodology and findings.

Introduction

It is well-written and informative. It provides a clear explanation of stable isotope ratios and their importance in studying biological processes. It also addresses the issue of accounting for lipids in stable isotope analysis, which is an important consideration that researchers need to take into account when interpreting results. The mention of the specific example of larval lampreys and the discrepancies in their stable isotope ratios adds an interesting layer to the discussion. However, I feel that few more recent works are available which can be incorporated.

We reconsidered every reference in the introduction but did not find any that we would want to replace. The work we are referencing is specific to larval lamprey isotopes, so while more work has been done on isotopes in lampreys, we have not included all isotopes studies with lampreys. If the reviewer feels we have missed a critical study, we would be happy to add it.

Discussion

Perfectly fine, and issues detected.

My general view is the paper can be accepted with slight changes as suggested.

We thank the reviewer for their time and effort.

---

## [Editor Report · Decision Letter 2]

18 May 2023

Are lipids always depleted? Comparison of hydrogen, carbon, and nitrogen isotopic values of lipids in larval lampreys

PONE-D-22-27112R2

Dear Dr. Evans

We’re pleased to inform you that your manuscript has been judged scientifically suitable for publication and will be formally accepted for publication once it meets all outstanding technical requirements.

Kind regards,

Dharmendra Kumar Meena

Academic Editor

PLOS ONE

Additional Editor Comments (optional):

Now author has addressed the comments so article ca be accepted.
---

## [Editor Report · Acceptance letter]

3 Jan 2024

PONE-D-22-27112R2 

PLOS ONE

Dear Dr. Evans, 

I'm pleased to inform you that your manuscript has been deemed suitable for publication in PLOS ONE. Congratulations! Your manuscript is now being handed over to our production team.

Kind regards, 

on behalf of

Dr. Dharmendra Kumar Meena 

Academic Editor

PLOS ONE